🔓 | **Open Peer Review** | Host-Microbial Interactions | Research Article

# The neutrophil oxidant hypothiocyanous acid causes a thiol-specific stress response and an oxidative shift of the bacillithiol redox potential in *Staphylococcus aureus*

Vu Van Loi,[1] Tobias Busche,[2] Franziska Schnaufer,[1] Jörn Kalinowski,[2] Haike Antelmann[1]

**ABSTRACT**   During infections, *Staphylococcus aureus* is exposed to hypochlorous acid (HOCl) and hypothiocyanous acid (HOSCN), which are produced by the neutrophil myeloperoxidase as potent antimicrobial killing agents. In this work, we applied RNAseq transcriptomics, Brx-roGFP2 biosensor measurements, and phenotype analyses to investigate the stress responses and defense mechanisms of *S. aureus* COL toward HOSCN stress. Based on the RNAseq transcriptome profile, HOSCN exerts strong thiol-specific oxidative, electrophile, and metal stress responses as well as protein damage in *S. aureus*, which is indicated by the strong induction of the HypR, TetR1, PerR, QsrR, MhqR, CstR, CsoR, CzrA, AgrA, HrcA, and CtsR regulons. Phenotype analyses of various mutants in HOSCN-responsive genes revealed that the HOSCN reductase MerA conferred the highest resistance toward HOSCN stress in *S. aureus* COL, whereas the QsrR and MhqR electrophile stress regulons do not contribute to protection. Brx-roGFP2 biosensor measurements and bacillithiol (BSH)-specific Western blot analyses revealed a strong oxidative shift of the bacillithiol redox potential ($E_{BSH}$) and increased *S*-bacillithiolations in *S. aureus*, indicating that BSH is oxidized to bacillithiol disulfide (BSSB) under HOSCN stress. While the Δ*merA* mutant was delayed in recovery of the reduced $E_{BSH}$, overproduction of MerA in the Δ*hypR* mutant enabled faster recovery of $E_{BSH}$ due to efficient HOSCN detoxification. Moreover, both MerA and BSH were shown to contribute to HOSCN resistance in growth assays. In summary, HOSCN provokes a thiol-specific oxidative, electrophile, and metal stress response, an oxidative shift in $E_{BSH}$ and increased *S*-bacillithiolation in *S. aureus*.

**IMPORTANCE**   *Staphylococcus aureus* colonizes the skin and the airways but can also lead to life-threatening systemic and chronic infections. During colonization and phagocytosis by immune cells, *S. aureus* encounters the thiol-reactive oxidant HOSCN. The understanding of the adaptation mechanisms of *S. aureus* toward HOSCN stress is important to identify novel drug targets to combat multi-resistant *S. aureus* isolates. As a defense mechanism, *S. aureus* uses the flavin disulfide reductase MerA, which functions as HOSCN reductase and protects against HOSCN stress. Moreover, MerA homologs have conserved functions in HOSCN detoxification in other bacteria, including intestinal and respiratory pathogens. In this work, we studied the comprehensive thiol-reactive mode of action of HOSCN and its effect on the reversible shift of the $E_{BSH}$ to discover new defense mechanisms against the neutrophil oxidant. These findings provide new leads for future drug design to fight the pathogen at the sites of colonization and infections.

**KEYWORDS**   *Staphylococcus aureus*, HOSCN, transcriptome, MerA, bacillithiol

Address correspondence to Haike Antelmann, haike.antelmann@fu-berlin.de.

The authors declare no conflict of interest.

See the funding table on p. 17.

*S*taphylococcus aureus is a commensal bacterium and part of the human skin and airway microbiota (1). At the same time, *S. aureus* can be a major human pathogen,

leading not only to local skin and soft-tissue infections but also to life-threatening systemic and chronic infections, e.g., septic shock, pneumonia, and osteomyelitis, especially in immunocompromised patients (2–4). Due to the high use of antibiotics in hospitals, the emergence of multi-resistant *S. aureus* (MRSA) isolates poses a high risk for treatment failure of *S. aureus* infections. Thus, the understanding of the adaptation and defense mechanisms of *S. aureus* during infections is an urgent need to identify new drug targets to combat life-threatening MRSA infections (5, 6).

After infection of the host, the pathogens are phagocytosed by the cells of our innate immune system, such as macrophages and neutrophils. The phagocytosis of the bacteria is associated with an oxidative burst of activated neutrophils, which is aimed to kill the invading bacteria (7–10). First, the NADPH oxidase is assembled in the phagosomal membrane, which transfers electrons to oxygen to produce reactive oxygen species (ROS), such as superoxide anions at concentrations of 2 mM/s (8–12). Upon fusion of the phagosomal membrane with azurophilic granules, the myeloperoxidase (MPO) is released into the phagosomal lumen. MPO dismutates superoxide to $H_2O_2$, which is used by MPO for the oxidation of chloride to produce the highly microbicidal hypochlorous acid (HOCl) (7–10). In the presence of the pseudohalide thiocyanate ($SCN^-$), MPO generates the hypothiocyanous acid (HOSCN), which is a similar powerful oxidant and antimicrobial agent as HOCl (7, 13). Although $SCN^-$ is the preferred substrate for MPO, the chloride outcompetes $SCN^-$ in the neutrophil phagosome, indicating that HOCl is the main oxidant produced by MPO (14). Apart from MPO, two other heme peroxidases, including lactoperoxidase (LPO) and eosinophil peroxidase (EPO) catalyze the reaction of $SCN^-$ with $H_2O_2$ to produce high levels of 1–3 mM HOSCN in the airways and saliva and 5–50 µM in the blood plasma of the human body (13, 15–17). Thus, *S. aureus* encounters HOSCN, especially at the sites of infections and colonization, including the airway mucosa of the upper respiratory tract where high $SCN^-$ levels are secreted (13).

While HOCl has been regarded as the most potent killing agent produced by neutrophils, its reaction with cellular macromolecules is rather non-specific, leading to thiol-oxidation and chlorination of proteins and subsequent protein aggregation as mechanisms of bacterial death (7, 13, 18, 19). In contrast, HOSCN is a more thiol-specific oxidant (20, 21), which reacts with protein thiols and low molecular weight (LMW) thiols, leading to the formation of unstable sulfenyl thiocyanate intermediates (R-S-SCN) and subsequently to protein disulfides, such as *S*-thiolations as well as intramolecular and intermolecular disulfides (7, 9, 10, 13, 22).

The bacterial responses and defense mechanisms against HOSCN stress have been investigated in important human pathogens, such as *Escherichia coli*, *Streptococcus pneumoniae*, *Pseudomonas aeruginosa*, and *S. aureus* (10, 13). While *P. aeruginosa* is highly sensitive toward HOSCN stress, the Gram-positive respiratory pathogens *S. pneumoniae* and *S. aureus* were found to be much more HOSCN resistant (23, 24). In buffer-killing assays, more than 50% of *S. aureus* and *S. pneumoniae* cells survived the treatment with 800 µM HOSCN after 2 hours, whereas *P. aeruginosa* was rapidly killed within 30 min (25). The HOSCN resistance of *S. pneumoniae* is mediated by the major LMW thiol glutathione (GSH), which is imported from the host via the GshT ABC transporter binding protein and maintained in the reduced state by the glutathione reductase Gor (24, 26–28). In addition, the flavin disulfide reductase (FDR) Har was identified as the main HOSCN reductase in *S. pneumoniae*, conferring GSH-independent resistance toward HOSCN (29), and deletion of both the GSH system and Har resulted in hypersensitivity toward HOSCN (29). Moreover, the Har homologous class-II FDR enzymes MerA and RclA were shown to confer strong resistance toward the neutrophil oxidants HOCl and HOSCN in *S. aureus* and *E. coli* (25, 30–32), supporting that MerA homologs represent the main defense mechanisms enabling survival during neutrophil infections.

Additionally, *S. aureus* encodes several antioxidant enzymes, such as the catalase KatA and the peroxiredoxin AhpCF, which play compensatory roles in the detoxification of the majority of $H_2O_2$ and confer high aerobic peroxide resistance (33, 34). The LMW thiol bacillithiol (BSH) and its associated bacilliredoxin (Brx)/BSH/bacillithiol disulfide

reductase (YpdA) pathway function in the defense against ROS and HOCl stress in *S. aureus* (35–38). Under HOCl stress, BSH is involved in widespread *S*-bacillithiolation of the thiol proteome, leading to thiol-protection and redox-regulation of redox-sensitive enzymes and transcription factors, such as the glyceraldehyde-3-phosphate dehydrogenase GapA of *S. aureus* and the OhrR repressor in *Bacillus subtilis* (39–42). The Brx/BSH/YpdA pathway is involved in the redox control and reversibility of *S*-bacillithiolations and is required for the survival of *S. aureus* during macrophage infections and HOCl stress (35, 36).

Furthermore, *S. aureus* encodes specific redox-sensing transcriptional regulators, such as PerR, MgrA, SarZ, HypR, QsrR, and MhqR, which respond to different redox-active compounds, e.g., ROS, HOCl, HOSCN, or reactive electrophile species by post-translational thiol-modifications, leading to conformational changes and inactivation or activation of the regulatory proteins (35, 43, 44). These redox regulators control adaptation and defense mechanisms, which are upregulated under specific redox stress conditions to degrade the redox-active species or to restore redox homeostasis (35, 43, 44). Recently, we have shown that the Rrf2-family repressor HypR senses neutrophil oxidants, such as HOCl and HOSCN stress, via Cys33-Cys99′ intersubunit disulfide formation, leading to inactivation of its repressor activity and upregulation of the HOSCN reductase MerA (25, 31). While the functions of MerA homologs toward HOSCN resistance have been explored in different pathogens (13, 23, 25, 29, 30), the genome-wide changes in gene expression and possible functions of other defense mechanisms and HOSCN-sensing redox regulators have not been systematically investigated in *S. aureus*.

In this study, we used global RNA-seq gene expression profiling to investigate the stress responses, which are upregulated under HOSCN, and to identify further defense mechanisms. HOSCN caused a strong thiol-specific oxidative, electrophile, and metal stress response as well as protein damage as revealed by the induction of the HypR, TetR1, PerR, QsrR, MhqR, CstR, CsoR, CzrA, AgrA, HrcA, and CtsR regulons. Thus, the gene expression profiles overlap strongly between HOCl and HOSCN stress, indicating similar adaptation and defense mechanisms in *S. aureus* against neutrophil oxidants. Using the Brx-roGFP2 biosensor, we further showed that HOSCN leads to a rapid reversible oxidative shift of the BSH redox potential ($E_{BSH}$), confirming its thiol-specific mode of action. The recovery of the reduced $E_{BSH}$ was impaired in the absence of MerA, whereas constitutive MerA expression in the $\Delta hypR$ mutant resulted in faster $E_{BSH}$ regeneration. Deletion of both MerA and BSH increased the susceptibility toward HOSCN stress during the growth, indicating that MerA and BSH confer independent HOSCN resistance. Thus, our study revealed the thiol-specific mode of action of HOSCN in *S. aureus* COL and the importance of MerA and the LMW thiol BSH in the defense against HOSCN stress.

## RESULTS

### *S. aureus* COL is more resistant toward HOSCN when grown in RPMI compared to LB medium

First, we determined the sub-lethal concentration of HOSCN, which impairs the growth of *S. aureus* COL, when cultivated in Luria-Bertani (LB) and RPMI medium, without killing effects. We used the LPO-SCN⁻ system to generate HOSCN, which was purified and quantified as described (23, 25, 31). *S. aureus* was grown to the mid-log phase and treated with increasing doses of 94–250 μM purified HOSCN to monitor the decrease of the growth rate (Fig. 1A and B). We noted that HOSCN has stronger inhibitory effects on the bacterial growth when *S. aureus* was cultivated in LB (Fig. 1A), whereas in RPMI medium even high concentrations of 250 μM HOSCN did not cause notable growth delays (Fig. 1B). When applied to the LB culture, the duration of the growth delay (60–120 min) increased with the HOSCN dose (136–250 μM), but even with 250 μM, the bacteria were able to succeed in growth after 2 hours with the same growth rate as the untreated cells. Thus, HOSCN detoxification must occur fast, allowing complete recovery of growth after the removal of HOSCN. For further transcriptome experiments, we have chosen

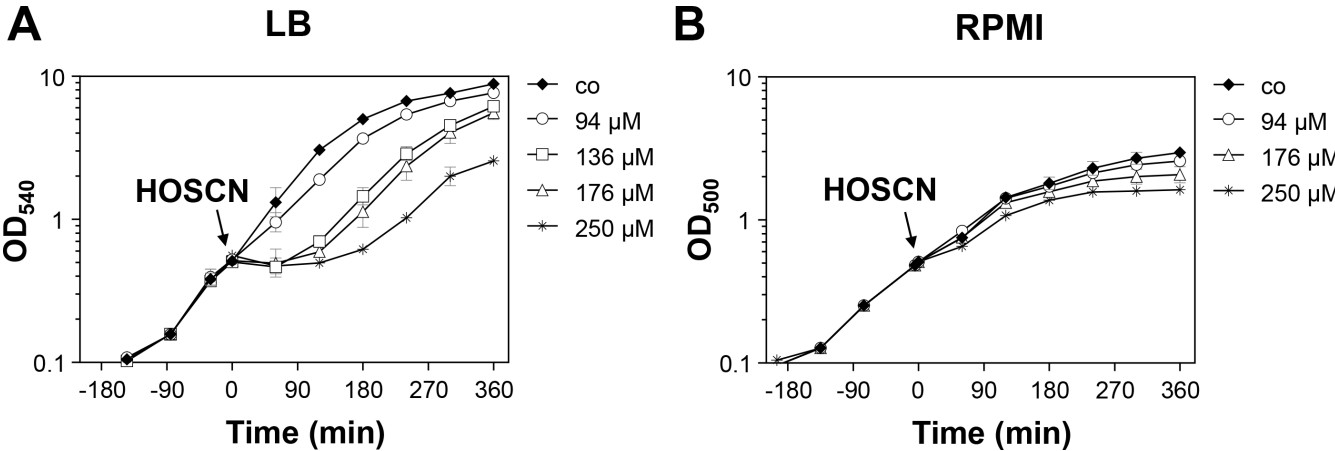

**FIG 1** Effect of HOSCN stress on the growth of *S. aureus* COL in LB (A) and RPMI medium (B). Growth curves of *S. aureus* COL in LB medium (A) and RPMI (B) were monitored after exposure to increasing doses of 94–250 µM HOSCN stress at an $OD_{540}$ and $OD_{500}$ of 0.5, respectively. *S. aureus* is more resistant toward HOSCN stress when grown in RPMI medium compared to LB. The results are from three biological replicate experiments. Error bars represent the standard deviation (SD).

176 µM HOSCN as a sublethal concentration in LB medium causing significant growth delays without killing effects in *S. aureus* cells.

## RNA-seq transcriptome analysis indicates a strong thiol-specific oxidative, electrophile, and metal stress response and protein damage under HOSCN stress in *S. aureus*

Next, we used RNA-seq transcriptome analysis to investigate the changes in gene expression in *S. aureus* COL, which was cultivated in LB in three biological replicates and exposed to 176 µM HOSCN for 30 min at an $OD_{540}$ of 0.5. To select genes with significant fold changes, we have chosen the *M*-value cutoff (log2-fold change HOSCN/control) of ≥1 and ≤−1. According to this cutoff, 649 and 702 genes were significantly more than twofold up- and downregulated, respectively, under HOSCN stress in *S. aureus* COL (Fig. 2; Tables S1 and S2).

Overall, the HOSCN transcriptome clearly indicates a characteristic thiol-specific oxidative stress signature, as observed also by other thiol-reactive compounds, including HOCl, allicin, and AGXX (31, 45, 46) (Tables S1 to S4). Among the top scorers were the 1,059–1,304-fold upregulated *SACOL2588-89* operon (TetR1 regulon) of unknown function and the HypR-controlled *hypR-merA* operon (77–154-fold), encoding the conserved HOSCN reductase MerA, conferring high resistance against HOSCN, HOCl, and allicin stress in *S. aureus* (25, 31, 46) (Fig. 2; Tables S1 and S2). In addition, the quinone-sensing QsrR and MhqR regulons respond strongly (7.5–38.7-fold) to HOSCN stress, including the *catE-SACOL0409-azoR1*, *yodC*, *catE2*, *frp*, and *mhqRED* operons, which function in the detoxification of quinones and confer resistance toward quinones, antibiotics, and oxidants (47, 48). The oxidative stress-specific PerR regulon, including the genes for antioxidant enzymes (*katA, ahpCF*, and *tpx*) and iron storage proteins (*dps* and *ftnA*), was strongly upregulated by HOSCN (3.4–40-fold) (49) (Fig. 2; Tables S1 and S2). Collectively, the strong induction of the HypR, TetR1, QsrR, MhqR, and PerR regulons revealed that HOSCN stress provokes a thiol-specific oxidative and electrophile stress response in *S. aureus*.

Among the HOSCN-responsive regulons were further the CtsR and HrcA regulons (13–270-fold), which control ATP-dependent chaperones (DnaK, GrpE, and GroESL) and proteases (ClpB, ClpC, and ClpP) to repair or degrade damaged proteins due to HOSCN-induced protein thiol-oxidation (50, 51) (Fig. 2; Tables S1 and S2). Additionally, HOSCN upregulates the transcription of the reactive sulfur species (RSS)-sensing CstR regulon, comprising the *cstAB-sqr* and *cstR-tauE* operons (3–33-fold), which encode the multi-domain sulfurtransferase (CstA), persulfide dioxygenase-sulfurtransferase (CstB), and

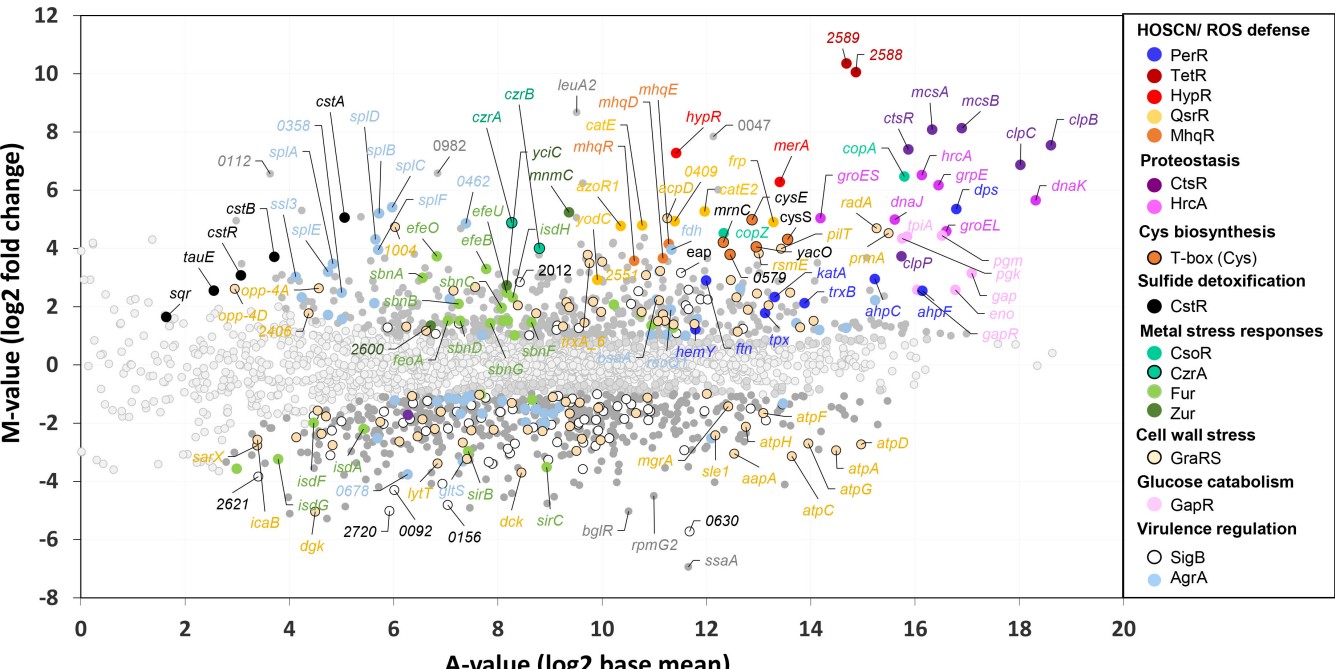

**FIG 2** RNA-seq transcriptomics indicates that HOSCN induces a thiol-specific stress response in *S. aureus* COL. For RNA-seq transcriptome profiling, *S. aureus* COL was grown in LB medium to an $OD_{540}$ of 0.5 and treated with sublethal 176 µM HOSCN stress for 30 min. The gene expression profile in response to HOSCN stress is shown as a ratio/intensity scatter plot (*M/A*-plot), which is based on the differential gene expression analysis using DeSeq2. The *M*-value represents the log2 fold change, and the *A*-value is the log2 average intensity (log2 base mean) of each transcript under HOSCN stress versus the untreated control. Light gray symbols denote transcripts with no fold changes ($P > 0.05$). Colored symbols and dark gray symbols indicate significantly induced or repressed transcripts (*M*-value ≥ 1.0 or ≤−1.0; $P ≤ 0.01$). Light gray symbols denote transcripts with no fold changes ($P > 0.05$). The significantly up- and downregulated regulons were functionally classified into the HOSCN/ROS defense (HypR, TetR, PerR, QsrR, and MhqR), proteostasis (CtsR, HrcA), sulfide detoxification (CstR), Cys biosynthesis (T-box Cys), metal stress responses (CsoR, CzrA, Fur, Zur), glucose catabolism (GapR), and virulence (SigB, AgrA) regulons, which revealed a strong thiol-specific oxidative and electrophile stress response and protein damage under HOSCN stress. These differentially expressed regulons with significant fold changes were color coded as indicated in the legend. The transcriptome analysis was performed from three biological replicates. The complete RNA-seq expression data of all genes after HOSCN stress and their regulon classifications are listed in Tables S1 and S2.

sulfide:quinone oxidoreductase (Sqr), involved in the detoxification of hydrogen sulfide in *S. aureus* (52–54). Thus, the sulfur compound HOSCN might resemble RSS since protein thiols are oxidized first to the intermediate sulfenyl thiocyanate (55–57). The thiol-specific oxidation by HOSCN further leads to the depletion of Cys and LMW thiols, indicated by the strong (13.8–31.5-fold) upregulation of the T-box Cys-controlled *cysE-cysS-mrnC-yacO-SACOL0579* operon, which is involved in the synthesis of O-acetyl-cysteine and cysteinyl-tRNA (58). In addition, the *cysK* gene encoding the cysteine synthase and *bshA* and *bshC* involved in BSH biosynthesis were 3.4–4.9-fold induced by HOSCN stress, indicating an enhanced requirement of Cys and BSH due to the thiol-specific oxidative mode of action of HOSCN (Fig. 2; Tables S1 and S2).

HOSCN causes the induction of the metal stress responsive CsoR and CzrA regulons (16–89-fold), controlling metal homeostasis by the upregulation of uptake systems for $Cu^+$ and efflux systems for $Zn^{2+}/Co^{2+}$, respectively (59–61) (Fig. 2; Tables S1 and S2). The metal-sensing repressors CsoR and CzrA share metal-binding Cys residues, which are most likely oxidized upon HOSCN, leading to the loss of their repressor activity and transcriptional upregulation of their regulons. In addition, some members of the Fur and Zur regulons, which respond to iron and $Zn^{2+}$ starvation, are weakly upregulated by HOSCN stress. Among the energy metabolism, the GapR-controlled *gapR-gap-pgk-tpiA-pgm eno* operon involved in glycolysis was significantly upregulated (6–21-fold), while operons encoding the pyruvate dehydrogenase and TCA cycle enzymes were not differentially transcribed (Fig. 2; Tables S1 and S2). Additionally, we noticed the

upregulation of the Rex-dependent pyruvate-formate lyase-encoding *pflAB* operon (3–10-fold) and the SrrAB-regulated *SACOL0219-hmp* operon (3.7–4.5-fold) encoding a flavohemoglobin under HOSCN stress, suggesting oxygen limitation and the switch to anaerobiosis.

Furthermore, the large GraRS cell wall stress regulon and the virulence regulons controlled by the accessory gene regulator A (AgrA) and the general stress and starvation sigma factor B (SigB) were differentially transcribed under HOSCN stress, with the majority of genes downregulated under HOSCN stress. Among the most strongly downregulated transcripts are the *atpCDGAHFEBI* operon encoding the ATP synthase (0.1–0.7-fold), the *sigB-rsbW-rsbV-rsbU* operon encoding the regulatory genes of SigB (0.13–0.16-fold), the *epiGEFPDCBA* operon for epidermin biosynthesis (0.1–0.5-fold), and the *crtNMQIO* operon for staphyloxanthin biosynthesis (0.06–0.27). In agreement with the HOCl-induced transcriptome changes, the *purEKCSQLFMNHD* operon for purine biosynthesis was partly downregulated (0.07–0.6-fold) under HOSCN stress, which could be due to limiting ATP levels since the ATP synthase operon was repressed (Fig. 2; Tables S1 and S2). However, while the PyrR and PurR regulons for pyrimidine and purine nucleotide biosynthesis and ribosomal operons involved in translation were strongly downregulated under HOCl stress, translation-associated genes were weakly 1.5–2.75-fold upregulated under HOSCN stress (Fig. 2; Tables S1 to S4). The downregulation of translation under stress and starvation conditions is associated with the synthesis of the alarmones (p)ppGpp, leading to a stringent response by the inhibition of processes required for active growth, while amino acid biosynthesis and stress response pathways are activated to promote bacterial survival (62). This lack of the stringent response by HOSCN stress might be due to the lower toxicity of HOSCN, which only causes reversible thiol-oxidation of proteins and LMW thiols, but cells are able to recover quickly in growth after detoxification of HOSCN (Fig. 1A).

Altogether, the HOSCN-specific transcriptome signature resembles that of other strong thiol-specific oxidants, such as HOCl, allicin, and AGXX stress, as revealed by the thiol-specific oxidative (HypR, PerR), electrophile (QsrR, MhqR), sulfur (CstR, T-box Cys), and metal stress response (CsoR, CzrA, Fur, Zur) as well as impaired proteostasis (HrcA, CtsR).

## Northern blot analyses verified the induction of the thiol-specific stress response by HOSCN stress in *S. aureus*

To confirm the RNA-seq expression data, we performed Northern blot analysis in *S. aureus* under HOSCN stress. We focused on the most strongly upregulated thiol-specific stress HypR, PerR, and QsrR regulons and the downregulated SigB regulon (Fig. 3A through C). Transcription of the HypR-controlled *merA* gene was most strongly 30-fold induced by HOSCN in Northern blots, followed by the 10–14-fold induction of the QsrR-regulated *azoR1* and *frp* transcripts. The PerR-dependent *dps*, *katA*, and *ahpCF* transcripts were similarly highly expressed under HOSCN stress, but *katA* and *ahpCF* had higher basal transcription levels, explaining their lower 3–3.5-fold induction ratios (Fig. 3A and B). The SigB-dependent genes *hchA* and *asp23* were strongly downregulated in the Northern blot analyses (Fig. 3A and C). Thus, the Northern blot data confirmed the strong induction of the thiol-specific stress response and downregulation of the SigB regulon under HOSCN stress as revealed by RNA-seq transcriptome data.

## The role of the HOSCN reductase MerA, antioxidant enzymes, and the bacillithiol system in the defense of *S. aureus* against HOSCN stress

Previously, we have shown that the NADPH-dependent flavin disulfide reductase MerA functions as HOSCN reductase and major defense mechanism against neutrophil oxidants in *S. aureus* USA300 JE2 (25, 31). In the RNA-seq transcriptome of *S. aureus* COL, the HypR, PerR, QsrR, and MhqR regulons were most strongly upregulated upon HOSCN stress, suggesting possible protective functions upon HOSCN stress (Fig. 2; Tables S1 and S2). Thus, we aimed to elucidate the contribution of the HypR-controlled MerA, the

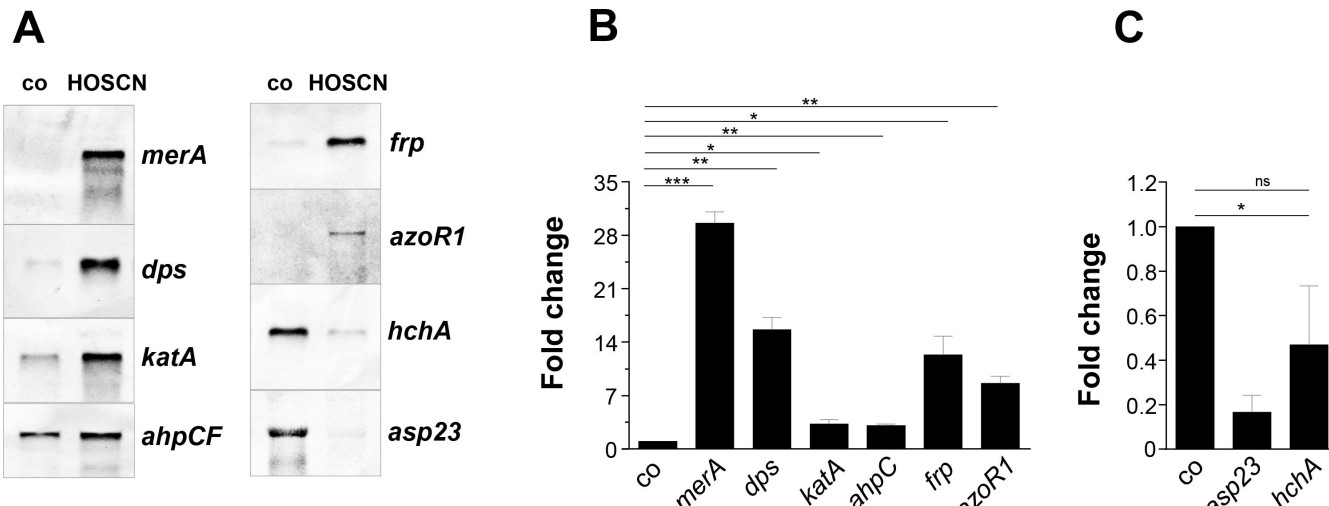

**FIG 3** Northern blot analysis confirmed the strong induction of the thiol-specific HypR, PerR, and QsrR regulons, while the SigB regulon is downregulated under HOSCN stress in *S. aureus*. (A) Transcription of the genes *merA* (HypR regulon), *dps, katA, ahpC* (PerR regulon), *frp, azoR1* (QsrR regulon), *hchA*, and *asp23* (SigB regulon) was analyzed in Northern blot experiments using RNA isolated from *S. aureus* COL WT 30 min after exposure to 176 µM HOSCN at an $OD_{540}$ of 0.5. (B and C) Quantification of the transcriptional induction of the genes after HOSCN stress in *S. aureus* was performed from the Northern blot images using ImageJ. HOSCN-induced fold changes were calculated from three biological replicates, and error bars represent the SD. For the calculation of the fold changes after HOSCN stress, the transcript intensities of each gene after HOSCN stress were normalized to the mRNA intensity of the untreated control, which was set to 1. The statistics of the fold changes for each gene under HOSCN stress versus the control were calculated using a Student's unpaired two-tailed *t*-test for two samples with unequal variance by the GraphPad Prism software. ns, $P > 0.05$; *$P \leq 0.05$; **$P \leq 0.01$; ***$P \leq 0.001$; and ****$P \leq 0.0001$.

PerR-dependent antioxidant response, the Brx/BSH/YpdA pathway, and the electrophile-sensing QsrR and MhqR regulons in the defense against HOSCN stress in *S. aureus* COL.

We analyzed the growth phenotypes of various mutants deficient in the thiol-specific oxidative and electrophile stress response, including the ΔmerA, ΔhypR, ΔperR, ΔkatA, ΔahpC, ΔkatAΔahpC, Δdps, ΔbrxAB, ΔbshA, ΔypdA, ΔqsrR, ΔmhqR, Δfrp, and ΔgbaA deletion mutants after exposure to sublethal doses of 176 µM HOSCN stress (Fig. 4; Fig. S1). The growth comparison of the *S. aureus* COL WT with the ΔmerA and ΔhypR mutants revealed an increased susceptibility of the ΔmerA mutant, while the ΔhypR mutant displayed increased resistance toward HOSCN due to the constitutive overexpression of MerA in the absence of the HypR repressor (31) (Fig. 4A and B). The mutant phenotypes could be restored to wild-type (WT) level in the *hypR*+ and *merA*+-complemented strains, leading even to better growth with HOSCN upon ectopic expression of *merA* from the xylose-inducible promoter of plasmid pRB473 (Fig. 4A and B). These results in the *S. aureus* COL background are consistent with the phenotypes of the *S. aureus* USA300 JE2 ΔmerA and ΔhypR mutants (25). However, systematic functional analyses of other single mutants deficient in the PerR-dependent antioxidant response, the Brx/BSH/YpdA redox system, or the QsrR and MhqR electrophile stress response did not show any significant growth phenotypes after HOSCN exposure in comparison to the WT (Fig. S1). This indicates that the HOSCN reductase MerA is the major HOSCN defense mechanism in *S. aureus*.

Previous analyses showed that the catalase KatA and the peroxiredoxin AhpCF could compensate for each other in $H_2O_2$ detoxification in *S. aureus* (33, 34). Indeed, the ΔkatAΔahpC double mutant was significantly impaired in growth upon HOSCN stress in relation to the WT, supporting that antioxidant enzymes protect against HOSCN-induced oxidative stress. Previous studies in *S. pneumoniae* revealed that the absence of both the MerA-homolog Har and the glutathione system leads to hypersensitivity toward HOSCN stress, whereas the Δhar single mutant did not display increased HOSCN sensitivity (29). Our growth analysis showed slightly enhanced sensitivity of the ΔmerAΔbshA double mutant versus the ΔmerA mutant upon exposure to 94 and 176 µM HOSCN stress,

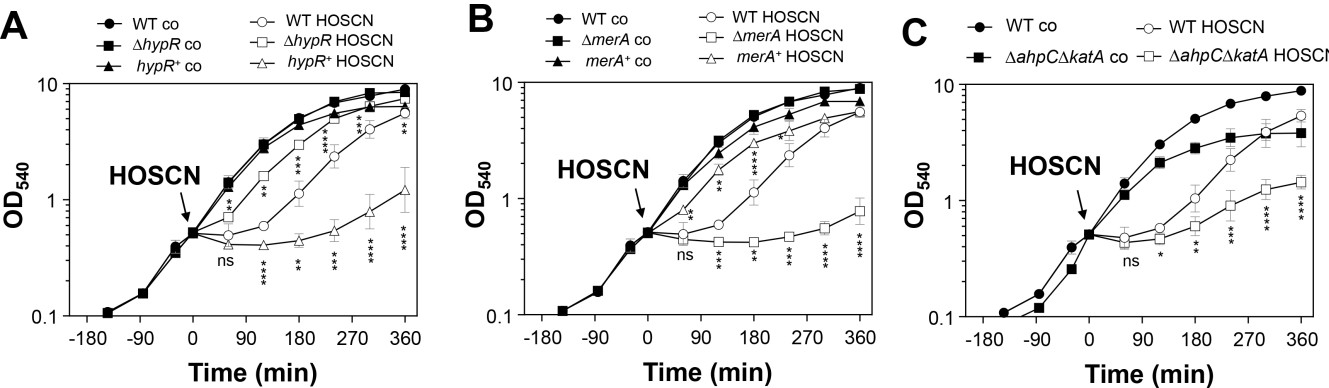

**FIG 4** The HOSCN reductase MerA plays a major role in the protection of *S. aureus* against HOSCN stress (A, B), and the absence of KatA and AhpC (C) also leads to HOSCN sensitivity. (A-C) The *S. aureus* COL WT, Δ*merA*, Δ*hypR*, and Δ*ahpC*Δ*katA* mutants and the *merA*[+] and *hypR*[+]-complemented strains were grown in LB until an $OD_{540}$ of 0.5 and treated with 176 µM HOSCN. Mean values were calculated from three biological replicates, and error bars represent the SD. The statistics of growth differences between the HOSCN-treated mutants or complemented strains versus the HOSCN-treated WT were calculated using a Student's unpaired two-tailed *t*-test for two samples with unequal variance by the GraphPad Prism software. ns, $P > 0.05$; *$P \leq 0.05$; **$P \leq 0.01$; ***$P \leq 0.001$; and ****$P \leq 0.0001$.

supporting that both MerA and the LMW thiol BSH contribute to HOSCN resistance (Fig. 5A and B).

Furthermore, viability assays of Δ*merA*, Δ*bshA,* and Δ*merA* Δ*bshA* mutants were conducted in the presence of high concentrations of 250 µM HOSCN, which caused a strong growth delay in the WT (Fig. 1A). Consistent with the growth phenotypes, the Δ*merA* mutant was significantly impaired in viability after 2 and 4 hours, whereas the Δ*bshA* mutant showed no viability defect compared to the WT (Fig. 5C). The *merA*[+]-complemented strain showed a strongly increased viability compared to the WT due to the constitutive expression of MerA from plasmid pRB473. However, the Δ*merA*Δ*bshA* double mutant was similarly impaired in viability as the Δ*merA* single mutant (Fig. 5C), indicating

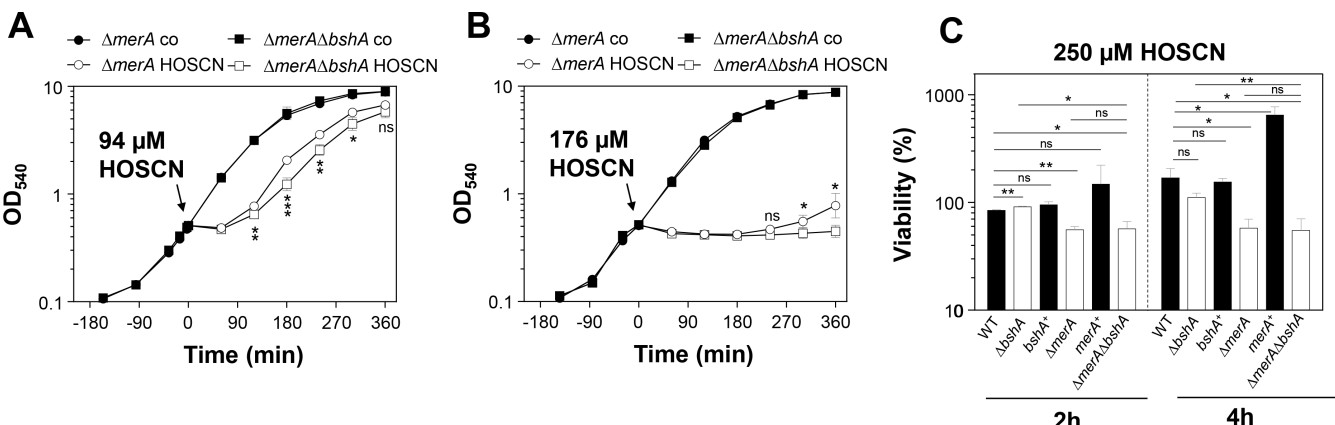

**FIG 5** In the absence of MerA, BSH significantly contributes to HOSCN resistance in growth assays but not in viability assays in *S. aureus*. (A and B) For growth curves, the *S. aureus* COL Δ*merA* and Δ*merA*Δ*bshA* mutants were grown in LB until an $OD_{540}$ of 0.5 and exposed to 94 or 176 µM HOSCN. Mean values were calculated from three biological replicates, and error bars represent the SD. The statistics of growth phenotypes were calculated between the HOSCN-treated Δ*merA*Δ*bshA* mutant versus the HOSCN-treated Δ*merA* mutant using a Student's unpaired two-tailed *t*-test for two samples with unequal variance by the GraphPad Prism software. (C) For viability assays, the *S. aureus* WT, Δ*bshA,* Δ*merA*, and Δ*merA*Δ*bshA* mutants and the *merA*[+]-complemented strain were treated with 250 µM HOSCN during the log phase and plated for CFUs after 2 and 4 hours of stress exposure. The percentage viability rate was calculated in relation to that of the untreated WT, which was set to 100%. Mean values were calculated from three biological replicates, and error bars represent the SD. The statistics of viability differences were calculated between the HOSCN-treated mutants or complemented strains versus the HOSCN-treated WT as well as between the HOSCN-treated Δ*merA*Δ*bshA* double mutant versus the Δ*merA* or Δ*bshA* mutants using a Student's unpaired two-tailed *t*-test for two samples with unequal variance by the GraphPad Prism software. ns, $P > 0.05$; *$P \leq 0.05$, **$P \leq 0.01$ and ***$P \leq 0.001$.

that BSH only makes a minor contribution toward HOSCN resistance to improve the growth of *S. aureus*. The susceptibility of the Δ*merA* mutant was also not affected in the Δ*merA*Δ*katA* or Δ*merA*Δ*frp* double deletion mutants, indicating that the catalase and the NADPH-dependent flavin reductase (Frp) cannot compensate for the absence of MerA in the defense against HOSCN stress (Fig. S2A through D).

Altogether, our phenotype analysis revealed that the HOSCN reductase MerA is the most important HOSCN defense mechanism of *S. aureus*, while the KatA/AhpCF antioxidant enzymes together provide some growth advantage under HOSCN stress. In addition, the LMW thiol BSH was shown to promote growth under HOSCN stress in the absence of MerA in *S. aureus*.

## The impact of MerA on the oxidation of the BSH redox potential under HOSCN stress in *S. aureus*

HOSCN acts as a thiol-specific oxidant, leading to reversible oxidation of protein thiols and LMW thiols, such as GSH and BSH (9, 13, 20, 21). Thus, we were interested in investigating whether HOSCN leads to the oxidation of BSH in *S. aureus* cells. Using the genetically encoded Brx-roGFP2 biosensor, the changes of the BSH redox potential ($E_{BSH}$) were measured as biosensor oxidation degree (OxD) inside *S. aureus* cells after HOSCN stress (Fig. 6A and B). The exposure of *S. aureus* WT to 94 and 176 µM HOSCN stress led to complete oxidation of the Brx-roGFP2 biosensor within 20 min, but the cells recovered very quickly to the fully reduced state of $E_{BSH}$ (Fig. 6A and B). The recovery phase was dose-dependent and required 60 and 100 min after the exposure to 94 and 176 µM HOSCN stress, respectively (Fig. 6A and B). These results revealed that HOSCN causes a strong reversible oxidative shift of the $E_{BSH}$ due to increased BSSB levels, which can be reduced by the BSSB reductase YpdA as shown previously (36).

To analyze the impact of HOSCN detoxification by MerA on the changes in $E_{BSH}$ upon HOSCN stress, the Brx-roGFP2 biosensor response was monitored in the Δ*merA* and Δ*hypR* mutant backgrounds (Fig. 6A and B). The Δ*merA* and Δ*hypR* mutants showed a similar fast oxidative shift of the $E_{BSH}$ upon HOSCN stress as the WT. However, the Δ*merA* mutant was significantly impaired in the regeneration of reduced $E_{BSH}$ after 176 µM HOSCN, whereas the Δ*hypR* mutant recovered much faster from HOSCN stress than the WT due to constitutive upregulation of MerA (Fig. 6A and B). These results strongly

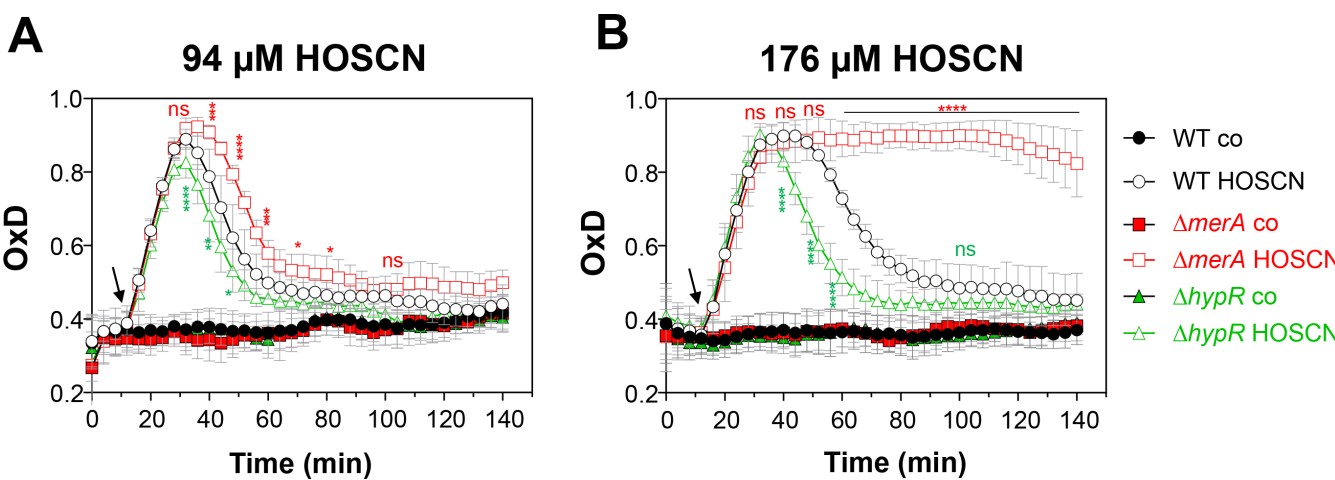

**FIG 6** HOSCN causes a strong oxidative shift in the BSH redox potential ($E_{BSH}$) as revealed by the Brx-roGFP2 biosensor. The *S. aureus* COL WT, Δ*merA,* and Δ*hypR* mutants expressing the Brx-roGFP2 biosensor were treated with 94 µM HOSCN (A) or 176 µM HOSCN (B) and the biosensor oxidation degree (OxD) was monitored using the CLARIOSTAR microplate reader as described (63). The Brx-roGFP2 biosensor shows a fast oxidative shift of $E_{BSH}$ upon HOSCN stress in all strains. The Δ*merA* mutant was delayed in the recovery of reduced $E_{BSH}$ after treatment with 176 µM HOSCN, while the recovery of the Δ*hypR* mutant was faster than the WT. Mean values and SD of three to four biological replicates are presented. The statistics of the OxD values were calculated for the HOSCN-treated mutants versus the HOSCN-treated WT at the 30-, 40-, 50-, 60-, 70-, 80-, and 100-min time points using a Student's unpaired two-tailed *t*-test for two samples with unequal variance by the GraphPad Prism software. ns, $P > 0.05$; *$P \leq 0.05$; **$P \leq 0.01$; ***$P \leq 0.001$; and ****$P \leq 0.0001$.

support the important role of MerA in HOSCN detoxification *in vivo*, and its contribution to the maintenance of cellular redox homeostasis. Since MerA also functions in the defense against HOCl stress (31), we analyzed next the Brx-roGFP2 biosensor response in the Δ*merA* and Δ*hypR* mutants after exposure to sublethal 100 µM HOCl stress. However, the Δ*merA* and Δ*hypR* mutants showed similar Brx-roGFP2 biosensor oxidation and recovery of reduced $E_{BSH}$ after exposure to 100 µM HOCl stress within 80 min as the WT (Fig. S3), indicating that MerA does not contribute alone to HOCl detoxification and the maintenance of reduced $E_{BSH}$ under HOCl stress.

## HOSCN causes increased *S*-bacillithiolations in the proteome of *S. aureus*

Previously, we showed that HOCl stress leads to strongly increased protein thiol-oxidation, including S-bacillithiolation of redox-sensitive proteins in *S. aureus* (40). Using BSH-specific non-reducing Western blot analysis, the glyceraldehyde dehydrogenase GapA was the most abundant *S*-bacillithiolated protein in *S. aureus* after HOCl stress (40). Thus, we applied non-reducing BSH-specific Western blot analysis to analyze the extent of protein *S*-bacillithiolation under HOSCN stress in *S. aureus*. Since MerA was shown to impact the recovery of $E_{BSH}$ after HOSCN stress (Fig. 6A and B), we analyzed the profile of *S*-bacillithiolations in the WT and Δ*merA* mutant after exposure to 176 µM HOSCN stress (Fig. 7A). The HOSCN treatment of the *S. aureus* WT and the Δ*merA* mutant resulted in increased protein *S*-bacillithiolations, but there was no difference between the WT and the Δ*merA* mutants, indicating that the deletion of *merA* does not affect protein *S*-bacillithiolations (Fig. 7A). The HOSCN-induced bands of *S*-bacillithiolations were not detected in the HOSCN-treated Δ*bshA* mutant and reversible in the WT with dithiothreitol (DTT) in the reducing Western blot analysis (Fig. 7A and B), confirming that HOSCN stress leads to increased protein *S*-bacillithiolations in the proteome of *S. aureus*. As noted earlier, the polyclonal BSH rabbit antiserum shows cross-reactivity with abundant cellular proteins (40), detected as background in the untreated WT and the Δ*merA* mutant as well as in the HOSCN-treated Δ*bshA* mutant. However, increased levels

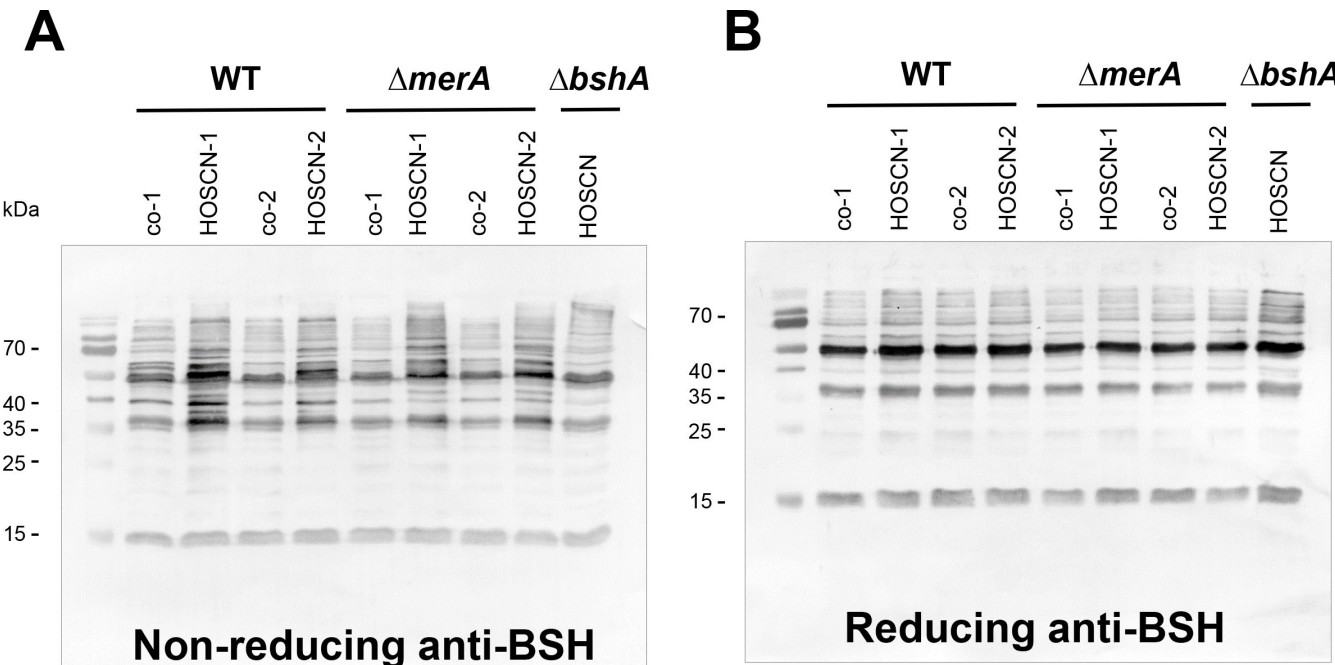

**FIG 7** HOSCN stress induces *S*-bacillithiolation in *S. aureus*. (A and B) The *S. aureus* COL WT, Δ*merA*, and Δ*bshA* mutants were exposed to 176 µM HOSCN for 30 min in BMM. The NEM-alkylated protein extracts were subjected to BSH-specific non-reducing and reducing Western blot analysis. HOSCN stress causes increased *S*-bacillithiolation in *S. aureus* WT and the Δ*merA* mutant. Two biological replicates are shown for the WT and the Δ*merA* mutant, denoted with co-1, co-2 and HOSCN-1, HOSCN-2, respectively.

of specific bands of *S*-bacillithiolated proteins were reproducibly detected in the WT and the Δ*merA* mutant under HOSCN stress. The specific targets for reversible thiol-oxidation including S-bacillithiolations under HOSCN stress will be elucidated using quantitative redox proteomics approaches in our future research.

## DISCUSSION

During infections and colonization of the upper respiratory tract, *S. aureus* encounters high concentrations of the pseudohypohalous acid HOSCN, which is generated by different host peroxidases, such as MPO, LPO, and EPO from $H_2O_2$ and $SCN^-$ in the airway, saliva, and plasma fluids (13, 15). Understanding the adaptation mechanisms of *S. aureus* toward HOSCN stress is important to tackle life-threatening MRSA infections and to improve human health. In this work, we have used RNA-seq transcriptomics to elucidate the global changes in gene expression of *S. aureus* upon HOSCN stress, identified new defense mechanisms, and the effect of HOSCN on the changes of the $E_{BSH}$ in *S. aureus* WT and mutants.

In the transcriptome, HOSCN stress provoked a strong thiol-specific oxidative, electrophile, and metal stress response as well as protein unfolding as revealed by the strong induction of the HypR, TetR1, PerR, QsrR, MhqR, CstR, CsoR, CzrA, AgrA, HrcA, and CtsR regulons. The oxidative and electrophile stress-specific HypR, TetR1, QsrR, and MhqR regulons were most strongly upregulated in the transcriptome under HOSCN, which clearly supports the thiol-reactive mode of action of HOSCN (Fig. 8). These redox regulons were also the top hits in the transcriptome of other thiol-reactive compounds,

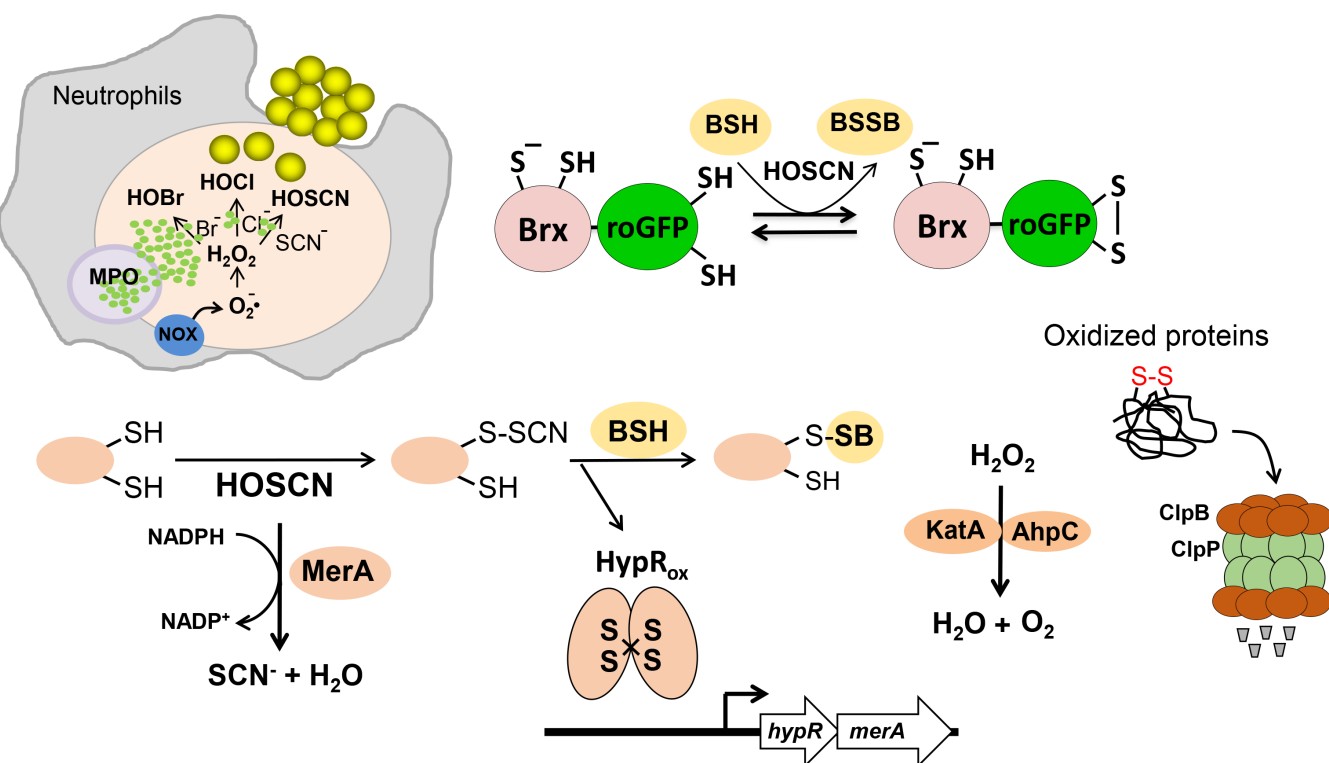

**FIG 8** The thiol-reactive mode of action of the neutrophil oxidants HOSCN in *S. aureus*. HOSCN is generated during infections inside neutrophils by the myeloperoxidase using $H_2O_2$ and $SCN^-$. HOSCN causes a strong thiol-specific oxidative, electrophile, and metal stress response and oxidative protein unfolding in *S. aureus* as revealed by the induction of the HypR, PerR, QsrR, MhqR, CzrA, CsoR, CtsR, and HrcA regulons in the transcriptome (Fig. 2). The HypR repressor is oxidized to HypR intermolecular disulfides by HOSCN stress, leading to its inactivation and upregulation of the flavin disulfide reductase MerA, which functions in HOSCN detoxification and contributes to the maintenance of the cellular redox homeostasis. Brx-roGFP2 biosensor measurements showed a reversible oxidative shift of the BSH redox potential upon HOSCN stress, which can be influenced by the expression level of MerA. In addition, the levels of *S*-bacillithiolated proteins increased after HOSCN stress. The changes in the cellular redox homeostasis cause ROS formation, which are detoxified by the catalase KatA and the peroxiredoxin AhpCF. The increased protein thiol-oxidation leads to protein unfolding, which requires the Clp proteases for degradation.

including HOCl, allicin, and AGXX stress (31, 45, 46). The detailed comparison of the gene expression changes upon HOCl and HOSCN stress (Table S4) further supports the strong overlap between the thiol-stress responses induced by both neutrophil oxidants.

The redox-sensing Rrf2-family HypR repressor controls the flavin disulfide reductase MerA, which confers resistance against HOSCN, HOCl, and allicin stress in *S. aureus* (25, 31, 46). The HypR repressor senses HOCl and HOSCN stress by intersubunit disulfide formation between Cys33 and Cys99' of opposing subunits, leading to inactivation of its repressor activity and derepression of transcription of the *hypR-merA* operon (25, 31) (Fig. 8). Similarly, the QsrR repressor senses quinones and oxidants via thiol-switch mechanisms *in vivo*, leading to induction of the dioxygenases and quinone reductases, which confer resistance against quinones and oxidants (48, 64). Under diamide stress, QsrR was reversibly oxidized to intersubunit disulfides between Cys4 and Cys29', while the organic sulfur compound allicin caused *S*-thioallylation of the Cys residues of QsrR, leading to its inactivation (48). Thus, the strong induction of the HypR and QsrR regulons supports the reversible thiol-oxidation of redox-sensing regulators under HOSCN stress.

Similarly, the redox-sensing RclR and NemR regulons were upregulated in *E. coli* and *P. aeruginosa* upon HOSCN and HOCl stress (32, 65–67), whereas the redox-sensitive NmlR regulon and the CTM electron complex (CcdA1, EtrxA1, and MsrAB2) for the repair of oxidized methionine residues (68, 69) were most strongly induced by HOCl in the transcriptome of *S. pneumoniae* (70). The NemR repressor of *E. coli* was shown to sense HOCl via thiol-switch mechanisms to control the expression of the glyoxalase GloA and the NEM reductase NemA, which confer resistance toward electrophiles and contribute to survival under HOCl stress (71–73). RclR of *E. coli* controls the *rclABC* operon, which encodes the MerA homologous flavin disulfide reductase RclA, which functions as highly efficient HOSCN reductase and protects *E. coli* against neutrophil oxidants, such as HOCl and HOSCN (30, 32, 74). In *P. aeruginosa*, the RclR homolog controls the peroxiredoxin RclX, which confers resistance toward HOSCN stress (65). However, *P. aeruginosa* was more sensitive toward HOSCN stress compared to other lung pathogens, such as *S. aureus* or *S. pneumoniae* (23). This higher HOSCN sensitivity of *P. aeruginosa* was proposed to be related to the absence of RclA/MerA/Har homologs, which are conserved in many bacteria and involved in HOSCN detoxification in *E. coli, S. aureus,* and *S. pneumoniae* (25, 29, 30, 75). Whether the RclR-dependent peroxiredoxin RclX of *P. aeruginosa* functions similarly efficiently in HOSCN detoxification remains to be investigated. In addition, many other thiol-dependent redox regulators, such as the metal-coordinating repressors CsoR, CzrA, Zur, and Fur, respond to both HOCl and HOSCN stress in *S. aureus* (31). Similarly, the SczA-regulated $Zn^{2+}$ efflux pump CzcD and the CopY-controlled copper-transporter CopA of *S. pneumoniae* were strongly upregulated under HOCl stress (70, 76, 77), and most likely respond also to HOSCN stress. Overall, there are parallels between the HOCl and HOSCN-induced redox stress regulons in different bacteria, governing the oxidative and electrophile stress responses in *E. coli*, *P. aeruginosa,* and *S. aureus* (67).

Moreover, HOSCN and HOCl stress causes protein thiol-oxidation and oxidative protein unfolding, leading to the strong upregulation of the CtsR and HrcA-regulons in *S. aureus* (31). The heat-shock-induced CtsR and HrcA repressors control chaperones and chaperonins (DnaK, GrpE, and GroESL) and the Clp proteases, which catalyze the ATP-dependent protein folding and degradation of unfolded proteins (50, 51) (Fig. 8). Similarly, chaperones and proteases controlled by the heat shock-specific sigma factor RpoH were upregulated in the transcriptome of *P. aeruginosa* after exposure to HOCl and HOSCN stress (65, 66), indicating related protein damage due to protein thiol-oxidation by both neutrophil oxidants. However, while protein aggregation is caused by HOCl stress as a bacterial killing mechanism (7, 18, 19), we did not observe the increased formation of protein aggregates in *S. aureus* after the exposure to 176 and 250 µM HOSCN stress (Fig. S4), which is consistent with the results obtained in *P. aeruginosa* WT cells after treatment with sub-lethal HOSCN stress (66).

We further showed that HOSCN exposure affects cellular redox homeostasis, leading to a strong oxidative shift in the $E_{BSH}$ of *S. aureus* as revealed by the genetically encoded Brx-roGFP2 biosensor. This indicates that HOSCN oxidizes the LMW thiol BSH to BSSB, supporting the thiol-reactive mode of action of HOSCN (Fig. 8). The recovery of the reduced state of $E_{BSH}$ after HOSCN exposure was dose-dependent, but fully reversible with 94 and 176 µM HOSCN. The biosensor oxidation profile was different between HOCl and HOSCN stress since *S. aureus* was unable to recover the reduced state upon higher doses of HOCl (63). Higher HOCl doses have been shown to cause overoxidation of BSH to form irreversible BSH sulfinic and sulfonic acids and BSH sulfonamide as revealed by thiol metabolomics (78). In contrast, HOSCN caused much lower levels of BSH sulfonamide than HOCl in *S. aureus* cells (78). Thus, the biosensor oxidation profiles after HOCl and HOSCN stress confirm the different thiol-reactive mode of action of both neutrophil oxidants. While HOSCN leads to reversible biosensor oxidation, higher HOCl concentrations induce irreversible oxidation products (78), which prevent the regeneration of the reduced state of the $E_{BSH}$ of *S. aureus* (63). The different growth behavior of *S. aureus* upon HOSCN and HOCl stress further supports their reversible and irreversible thiol-oxidation capabilities (Fig. 1A; Fig. S5). While higher doses of HOCl do not lead to the recovery of growth, *S. aureus* is able to resume the growth after detoxification of high doses of HOSCN (Fig. 1A; Fig. S5) (25). The depletion of the LMW thiol BSH by HOSCN and HOCl stress results in strong induction of Cys and BSH biosynthesis genes in the transcriptome of *S. aureus*, including the T-box Cys-controlled *cysE-cysS-mrnC-yacO-SACOL0579* operon, *cysK*, *bshA*, and *bshC*. The increased transcription of genes for the biosynthesis of Cys and methionine was also observed upon exposure to the neutrophil oxidants in *P. aeruginosa*, supporting thiol depletion (65, 66). Furthermore, BSH-specific Western blot analyses confirmed the increased level of protein *S*-bacillithiolations under both HOCl and HOSCN stress in *S. aureus* (40). In future studies, we aim to conduct a more detailed redox proteomics analysis to determine the percentage of increased reversible thiol-oxidation upon HOSCN stress.

Based on the transcriptome, we aimed to discover novel HOSCN defense mechanisms of *S. aureus*. Thus, we analyzed the growth phenotypes of various single mutants involved in the oxidative and electrophile stress response, including the Brx/BSH/YpdA redox system, the peroxide-specific PerR regulon (KatA, AhpC, and Dps), and the QsrR, MhqR, and GbaA electrophile stress regulons. However, we did not detect any growth phenotype of the single mutants after exposure to HOSCN stress. Only the HypR-controlled HOSCN reductase MerA was important for growth under HOSCN stress in *S. aureus* COL, confirming our previous results in the *S. aureus* USA300 JE2 background (25). The *S. aureus* COL Δ*merA* mutant was highly sensitive under HOSCN stress, whereas the Δ*hypR* mutant and the *merA*⁺-complemented strain showed constitutive resistance toward HOSCN due to the overexpression of MerA. Thus, the HOSCN reductase MerA was revealed as a major defense mechanism in *S. aureus* COL. In this study, we further showed that MerA contributed to the regeneration of the reduced state of the $E_{BSH}$ after recovery from HOSCN stress. The regeneration of the reduced state of the $E_{BSH}$ was impaired in the absence of MerA, while constitutive MerA expression in the Δ*hypR* mutant resulted in faster recovery of reduced $E_{BSH}$, clearly indicating that MerA is involved in HOSCN detoxification *in vivo*. Apart from the Δ*merA* mutant, only the Δ*katA*Δ*ahpC* double mutant was sensitive toward HOSCN stress. However, the absence of KatA and AhpC affects aerobic growth due to the hypersensitivity of the Δ*katA*Δ*ahpC* mutant toward $H_2O_2$ stress (33, 34). Since overexpression of KatA and AhpC in the Δ*perR* mutant did not provide a growth advantage under HOSCN stress, the role of the antioxidant enzymes KatA and AhpC in protection against HOSCN-induced ROS formation remains unclear.

Previously, the roles of the LMW thiol GSH and the MerA homolog Har of *S. pneumoniae* have been investigated in the defense against HOSCN stress (24, 29). *S. pneumoniae* does not encode the enzymes for GSH biosynthesis and instead imports GSH from the host using the ABC transporter-binding protein GshT, which is important for growth and oxidative stress resistance (26–28). To keep host-derived GSH in a reduced state, *S.*

*pneumoniae* encodes the GSSG reductase Gor (28). HOSCN leads to the oxidation of GSH to GSSG and the increased formation of S-glutathionylated proteins in *S. pneumoniae* (24). While the Δ*har* mutant did not show increased HOSCN susceptibility (29), deletion of either GshT or Gor resulted in impaired growth after HOSCN stress (24). However, the Δ*har*Δ*gshT* and Δ*har*Δ*gor* double mutants were hypersensitive toward HOSCN compared to the Δ*gor* and Δ*gshT* single mutants (29). This indicates that the GSH/Gor system and Har play compensatory roles in the defense against HOSCN stress in *S. pneumoniae* (29).

In our study, deletion of the Brx/BSH/YpdA system alone did not cause any growth defect under HOSCN, whereas the absence of both MerA and BSH led to slightly reduced growth rates under HOSCN stress compared to the Δ*merA* single mutant. However, there was no viability difference between Δ*merA* single and Δ*merA*Δ*bshA* double mutants, indicating that BSH only contributes to improved growth under HOSCN stress in the absence of MerA. These results indicate that MerA is more important than the Brx/BSH/YpdA system in protection against HOSCN in *S. aureus*. In contrast, GSH import seems to be more essential for the defense against HOSCN in *S. pneumoniae*, whereas the contribution of Har to HOSCN detoxification is only relevant in the absence of GSH import or recycling (24, 29). Furthermore, while RclA and MerA both respond strongly to HOSCN and HOCl stress (25, 30–32), the homolog Har was not upregulated under HOCl and HOSCN stress in *S. pneumoniae* (29, 70). Thus, the lack of phenotype of the Δ*har* single mutant toward HOSCN stress might be related to the fact that Har is not inducible by neutrophil oxidants. Future analyses should investigate in more detail the transcriptome changes upon HOCl and HOSCN stress and the redox-sensing mechanisms involved in the defense against neutrophil oxidants in other pathogenic or colonizing bacteria.

## MATERIALS AND METHODS

### Bacterial strains and growth conditions

The bacterial strains, plasmids, and primers used in this study are described in Tables S5 and S6. For genetic manipulation, *E. coli* strains were grown in LB medium. While previous phenotype analyses were performed with the *S. aureus* USA300 JE2 wild type, Δ*hypR,* and Δ*merA* mutants, we have used *S. aureus* COL in this work, due to the availability of a large mutant collection constructed in the COL background. *S. aureus* USA300 JE2 is a community-acquired highly virulent MRSA isolate, cured of two antibiotic-resistance plasmids (79). *S. aureus* COL is an archaic hospital-acquired MRSA isolate of lower virulence compared to USA300 JE2 (80). The *S. aureus* COL strains were cultivated in LB, Belitsky minimal medium (BMM), or RPMI supplemented with 0.75 μM $FeCl_2$ and 2 mM glutamine as described (81). For the growth of the *S. aureus*-complemented strains harboring the pRB473-plasmids, the supplementation of the medium with 1% xylose was required for the expression of the protein. For stress experiments, *S. aureus* strains were treated with sublethal HOSCN concentrations at an optical density at 540 nm ($OD_{540}$) of 0.5 as described previously (25). Viability assays were performed by plating 100 μL of serial dilutions of *S. aureus* cultures after treatment with the lethal dose of 250 μM HOSCN on LB agar plates for counting of colony forming units (CFUs). If appropriate, antibiotics were used at the following concentrations: 100 μg/mL ampicillin, 10 μg/mL erythromycin, and 10 μg/mL chloramphenicol. HOSCN was generated using the LPO-SCN⁻ system, where $H_2O_2$ and SCN⁻ were converted by the LPO to HOSCN according to previous protocols (23, 25, 31). The compounds sodium hypochlorite (NaOCl), diamide, DTT, and N-ethyl maleimide (NEM) were purchased from Sigma Aldrich. Statistical analysis of the growth curves, viability assays, Northern blot transcription, and Brx-roGFP2 biosensor measurements was performed using Student's unpaired two-tailed *t*-test for two samples with unequal variance by the GraphPad Prism software. The specific samples compared in the Student's unpaired two-tailed *t*-test are indicated in the figure legends and the *P*-values are shown for each comparison.

## RNA isolation, library preparation, and next-generation cDNA sequencing

For the HOSCN transcriptome analysis, *S. aureus* COL was grown in LB medium to the exponential phase of an $OD_{540}$ of 0.5 and exposed to sublethal 176 µM HOSCN, which was generated freshly using the $LPO-SCN^-$ system. The HOCl transcriptome data were generated from *S. aureus* USA300, which was grown in RPMI to an $OD_{500}$ of 0.5 and exposed to 1.5 mM HOCl for 30 min. *S. aureus* cells were harvested before (as control) and 30 min after exposure to HOSCN or HOCl stress. Subsequently, the cells were disrupted in 3 mM ethylenediaminetetraacetic acid (EDTA)/200 mM NaCl lysis buffer using a Precellys24 ribolyzer. RNA isolation was performed using the acid phenol extraction protocol as described (82). The RNA quality was checked by Trinean Xpose (Gentbrugge, Belgium) and the Agilent RNA Nano 6000 kit using an Agilent 2100 Bioanalyzer (Agilent Technologies, Böblingen, Germany). Ribo-Zero rRNA Removal Kit (Bacteria) from Illumina (San Diego, CA, USA) was used to remove the rRNA. TruSeq Stranded mRNA Library Prep Kit from Illumina was applied to prepare the cDNA libraries. The cDNAs were sequenced paired end on an Illumina NextSeq 500 system (San Diego, CA, USA) using 75 bp read length. The transcriptome sequencing raw data files are available in the ArrayExpress database under accession number E-MTAB-13314 for the HOSCN stress data set and as E-MTAB-13313 for the HOCl stress data set used for comparison.

## Bioinformatics data analysis, read mapping, data visualization, and analysis of differential gene expression

The paired-end cDNA reads were mapped to the *S. aureus* COL genome sequence (accession number NC_002951) using bowtie2 v2.2.7 (83) with default settings for paired-end read mapping. All mapped sequence data were converted from SAM to BAM format with SAMtools v1.3 (84) and imported to the software ReadXplorer v.2.2 (85).

Differential gene expression analysis of triplicates including normalization was performed using Bioconductor package DESeq2 (86) included in the ReadXplorer v2.2 software (85). The signal intensity value (*A*-value) was calculated by the log2 mean of normalized read counts and the signal intensity ratio (*M*-value) by the log2 fold change. The evaluation of the differential RNA-seq data was performed using an adjusted *P*-value cutoff of $P \leq 0.01$ and a signal intensity ratio (*M*-value) cutoff of $\geq 1$ or $\leq -1$. Genes with an *M*-value outside this range and $P \leq 0.05$ were considered as differentially up- or downregulated under HOSCN stress.

## Construction of the *S. aureus* COL Δ*frp*, Δ*dps*, Δ*merA*Δ*bshA*, Δ*merA*Δ*frp,* and Δ*merA*Δ*katA* mutants and the Brx-roGFP2 biosensor strains in the Δ*merA* and Δ*hypR* mutant backgrounds

The *S. aureus* COL Δ*frp* (*SACOL2534*) and Δ*dps* (*SACOL2131*) single and the Δ*merA*Δ*bshA*,Δ*merA*Δ*katA* and Δ*merA*Δ*frp* double deletion mutants were constructed using the temperature-sensitive shuttle vector pMAD as described (31). For the Δ*dps* and Δ*frp* single mutants, the 500 bp up- and downstream regions of the coding regions were amplified using gene-specific primers (Table S6). Subsequently, the PCR products were fused by overlap extension PCR and ligated into the *Bgl*II and *Sal*I sites of plasmid pMAD. The obtained pMAD constructs were electroporated into the intermediate *S. aureus* RN4220 strain for methylation and further transduced into *S. aureus* COL using phage 81 (87). The clean marker-less Δ*frp* and Δ*dps* deletion mutants were selected after plasmid excision as described (31). The Δ*merA*Δ*bshA*,Δ*merA*Δ*katA* and Δ*merA*Δ*frp* double mutants were obtained by transducing phage 81, which carried the plasmids pMAD-Δ*bshA*, pMAD-Δ*katA,* or pMAD-Δ*frp* into the *S. aureus* COL Δ*merA* mutant. Selection of the double deletion mutants was performed as previously described (31). The deletions of internal gene regions were confirmed by PCR and DNA sequencing (31).

The *S. aureus* COL Δ*hypR*-pRB473-*brx-roGFP2* andΔ*merA* pRB473-*brx-roGFP2* biosensor strains were constructed by transduction of the plasmid pRB473-*brx-roGFP2* into the *S. aureus* COL Δ*hypR* and Δ*merA* mutants using the phage 81 (87). The pRB473-*brx-roGFP2*

biosensor was confirmed in the ΔmerA and ΔhypR mutants by PCR and fluorescence microscopy.

## Northern blot analysis

Northern blot transcriptional analysis was performed using RNA isolated from *S. aureus* COL before and 30 min after exposure to 176 µM HOSCN as described (82). Hybridizations were conducted using digoxigenin-labeled antisense RNA probes for *frp* and *hchA* that were synthesized *in vitro* using T7 RNA polymerase as in previous studies (31). The RNA probes specific for the genes *merA, katA, ahpC, dps, azoR1,* and *asp23* were generated previously (31, 46, 88).

## Western blot analysis to analyze *S*-bacillithiolations

To analyze protein *S*-bacillithiolations after HOSCN stress, non-reducing BSH-specific Western blot analysis of *S. aureus* COL WT strain and the ΔmerA and ΔbshA mutants after exposure to HOSCN was conducted as previously described (31). The *S. aureus* cells were cultivated in LB medium until an $OD_{540}$ of 2, transferred to BMM, and treated with 176 µM HOSCN for 30 min. Cells were collected in the presence of 50 mM NEM, washed in Tris-EDTA (TE) buffer (pH 8.0), and disrupted using the Precellys24 ribolyzer. Protein amounts of 25 µg were separated under non-reducing conditions using 15% SDS-PAGE and the Western blot analysis was performed as described previously (39). For control purposes, the Western blot analysis was performed with the same samples under reducing conditions (with DTT) to reduce oxidized proteins with reversible *S*-bacillithiolations. Polyclonal rabbit anti-BSH antiserum was used at a dilution of 1:500 for Western blot analyses.

## Brx-roGFP2 biosensor measurement to analyze the changes in the BSH redox potential ($E_{BSH}$) in *S. aureus* COL

The *S. aureus* COL WT, ΔhypR, and ΔmerA mutant strains expressing the Brx-roGFP2 biosensor were grown in LB medium as overnight culture, transferred to BMM, and adjusted to an $OD_{500}$ of 3. Subsequently, the bacterial cells were transferred to the microplates and the biosensor OxD was monitored after the injection of HOSCN as described previously (63, 89). To obtain the fully reduced and oxidized control samples, the *S. aureus* COL Brx-roGFP2 expression strains were exposed to 10 mM DTT and 10 mM diamide, respectively. The Brx-roGFP2 biosensor fluorescence emission was measured at 510 nm after excitation at 405 and 488 nm using the CLARIOstar microplate reader (BMG Labtech). The time-dependent changes of the biosensor OxD were measured continuously for each sample and normalized to the fully reduced and oxidized controls as described (63, 89).

## Isolation of protein aggregates after HOSCN stress

Intracellular protein aggregates were isolated from *S. aureus* cells, which were grown in LB medium and exposed to 176 and 250 µM HOSCN stress for 30 min at an $OD_{540}$ of 0.5, as described in previous protocols (45, 66, 90). In brief, the cells were resuspended in 40 µL buffer A (10 mM potassium phosphate pH 6.5, 1 mM EDTA, 20% sucrose, 5 µg/mL lysostaphin) and 360 µL buffer B (10 mM potassium phosphate pH 6.5, 1 mM EDTA), followed by disruption using the ribolyzer. Subsequently, the insoluble protein aggregates were collected and analyzed using SDS-PAGE as described (45).

## ACKNOWLEDGMENTS

This work was supported by grants from the Deutsche Forschungsgemeinschaft, Germany (AN746/4-1 and AN746/4-2) within the SPP1710 on "Thiol-based Redox switches," by the SFB973 (project C08) and TR84 (project B06) to H.A.

No competing financial interests exist.

## AUTHOR AFFILIATIONS

[1]Institute of Biology-Microbiology, Freie Universität Berlin, Berlin, Germany

[2]Microbial Genomics and Biotechnology, Center for Biotechnology, Bielefeld University, Bielefeld, Germany

## AUTHOR ORCIDs

Haike Antelmann http://orcid.org/0000-0002-1766-4386

## FUNDING

| Funder | Grant(s) | Author(s) |
| --- | --- | --- |
| Deutsche Forschungsgemeinschaft (DFG) | AN746/4-1, AN746/4-2, SFB973/C08, TR84/B06 | Haike Antelmann |

## AUTHOR CONTRIBUTIONS

Vu Van Loi, Data curation, Formal analysis, Investigation, Methodology, Software, Writing – review and editing | Tobias Busche, Data curation, Formal analysis, Methodology, Resources, Software, Writing – review and editing | Franziska Schnaufer, Formal analysis, Investigation, Methodology, Writing – review and editing | Jörn Kalinowski, Data curation, Formal analysis, Methodology, Resources, Software, Writing – review and editing | Haike Antelmann, Conceptualization, Funding acquisition, Investigation, Project administration, Resources, Supervision, Validation, Visualization, Writing – original draft, Writing – review and editing

## DATA AVAILABILITY

The authors confirm that the data supporting the findings of this study are available within the article and its supplemental materials. The transcriptome sequencing raw data files are available in the ArrayExpress database under accession number E-MTAB-13314 for the HOSCN stress data set and as E-MTAB-13313 for the HOCl stress data set used for comparison.

## ADDITIONAL FILES

The following material is available online.

### Supplemental Material

**Fig. S1 to S5 (Spectrum03252-23-S0001.pdf).** Growth curves for phenotype analyses, Brx-roGFP2 biosensor experiments, and aggregation assays.
**Tables S1 to S4 (Spectrum03252-23-S0002.xlsx).** Transcriptome datasets of *S. aureus* after exposure to HOSCN and HOCl stress.
**Tables S5 and S6 (Spectrum03252-23-S0003.pdf).** Bacterial strains, phages, plasmids, and oligonucleotide primers.

### Open Peer Review

**PEER REVIEW HISTORY (review-history.pdf).** An accounting of the reviewer comments and feedback.

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
