## [Reviewer comments · Microbiology Spectrum]

Microbiology Spectrum

The neutrophil oxidant hypothiocyanous acid causes a thiol-specific stress response and an oxidative shift of the bacillithiol redox potential in *Staphylococcus aureus*

Vu Loi, Tobias Busche, Franziska Schnauffer, Jörn Kalinowski, and Haike Antelmann

Corresponding Author(s): Haike Antelmann, Freie Universität Berlin

Review Timeline:

Submission Date:	September 1, 2023
Editorial Decision:	September 22, 2023
Revision Received:	September 29, 2023
Accepted:	October 2, 2023

Editor: Artem Rogovskyy

Reviewer(s): The reviewers have opted to remain anonymous.

Transaction Report:

DOI: <https://doi.org/10.1128/spectrum.03252-23>

September 22, 2023

Prof. Haike Antelmann
Freie Universität Berlin
Institute of Biology-Microbiology
Königin-Luise-Str. 12-16
Berlin D-14195
Germany

Re: Spectrum03252-23 (The neutrophil oxidant hypothiocyanous acid causes a thiol-specific stress response and an oxidative shift of the bacillithiol redox potential in *Staphylococcus aureus*)

Dear Prof. Haike Antelmann:

Link Not Available

Sincerely,

Artem Rogovskyy

Journals Department
Reviewer comments:

Reviewer #1 (Comments for the Author):

Review of Spectrum03252-23: "The neutrophil oxidant hypothiocyanous acid causes a thiol-specific stress response and an oxidative shift of the bacillithiol redox potential in *Staphylococcus aureus*"

Summary:

In this thorough and informative manuscript, Loi and colleagues dissect the response of the important pathogen *Staphylococcus aureus* to the immune oxidant hypothiocyanous acid (HOSCN), an innate immune antimicrobial for which bacterial defenses

have only recently begun to be characterized. They compare the impact of HOSCN and the better-studied oxidant HOCl on the transcriptome of *S. aureus*, as well as examining the role of diverse antioxidant gene systems and the low-molecular weight thiol bacillithiol in the HOSCN response, coming to the conclusion that in this organism the HOSCN reductase MerA is the primary defense mechanism against HOSCN, expanding on the conclusions of their previous paper in this area.

On the whole, this is an excellent paper and a valuable contribution to the literature. There are only a few points of confusion and suggestions that I have, which are detailed below.

Specific Comments:

General point: This paper uses *S. aureus* strain COL, as opposed to their previous work on strain USA300 JE2. Is there an advantage to this difference? It might be useful to mention at some point in the manuscript the relevant differences between these strains.

Figures 3-6: Student's t-test is not appropriate for analyzing these data. Please consult a statistician to select the most appropriate analysis, but I believe some form of ANOVA with a multiple comparisons test would be required for each of these figures.

Lines 145-158: What is the relevance or interpretation of increased resistance to HOSCN in RPMI vs. LB? Does RPMI contain more HOSCN-reactive thiols than LB?

Line 152: In this paper, 250 μ M HOSCN is referred to as a "high" HOSCN concentration. In papers published on other species (e.g. *S. pneumoniae* or *E. coli*), HOSCN doses as high as 800 μ M are common. How does the ability of *S. aureus* to resist HOSCN compare to other species? *S. pneumoniae* may be the most relevant comparison here, since they are both Gram-positive organisms encoding MerA/Har.

Lines 210-214: I find the report of down-regulation as fractional values (e.g. "0.1-fold") a little confusing and counter-intuitive. I would much prefer that this same difference be expressed as a 10-fold reduction in expression instead (and similarly for all of the fractions in this section).

Line 219: For readers not familiar with the stringent response, it would be useful to define that before this point.

Line 271: I don't know if I would characterize this as "similarly", especially since the authors note in the Discussion how GSH seems to be much more important to HOSCN response in *S. pneumoniae* than BSH is in *S. aureus*.

Lines 275-286 and Figure 5C: I do not think this qualifies as a "% survival assay" when most of the strains reported are either not dying at all or are growing more or less robustly. It might be better to simply refer to this as a viable cell assay. Do the colors of the bars have any meaning? I suggest plotting the percent cell viability on the Y-axis on a log scale, which would have the advantage of making the 100% value more obvious (a dashed line there would also help the reader tell whether any of the WT cells, for example, are being killed) and would also remove the need for a broken axis. In general, I prefer plotting individual points rather than or in addition to histogram bars, but I recognize that is a personal aesthetic choice.

Figure 2: This figure is extremely complex, and while there may be a limit to what the authors can do about that (the data are complex!), I have a few suggestions that may help. While A-value is defined in the Methods section, it is not obvious from what's in the figure and legend. It may be better to express this in simpler terms for the non-expert; perhaps "A-value (log₂ mean gene expression)"? A definition in the figure legend is certainly necessary. Secondly, I don't think the difference between light grey and dark grey dots is explained anywhere. I think the dark grey with black borders are the SigB operon, but in general, I'm struggling with the multiple uses of grey in this figure. I recognize the challenge of finding a color scheme that works well in a figure like this, but I have found that the palettes described at <https://personal.sron.nl/~pault/> are helpful in this regard.

Figure 3B: I will argue again for a log-scale rather than a broken axis for this graph as well.

Figure 7: This is the figure I have the most trouble with in the paper. If this Western blot uses antibodies specific to protein bacillithiolation, why are there any bands detected in the Δ bshA strain? There are certainly more bands in the lanes with HOSCN, and those do appear to be eliminated in the reducing gel, but I'm very puzzled by the presence of so many dominant bands that are neither BSH- nor HOSCN-dependent. Please include more information on the interpretation of this experiment.

Reviewer #2 (Comments for the Author):

In the current manuscript, the authors follow up on their previous findings and investigate how the gram-positive bacterium *S. aureus* responds to hypothiocyanous acid stress using transcriptomic, phenotypic, and biosensor approaches. The manuscript is well-written and easy to follow. While the growth curve analyses of Δ merA and Δ hypR strains during HOSCN stress are not novel (published in Ref 29 and acknowledged by the authors in lines 101-105), a transcriptomic response and the cellular

oxidation status have not been reported before. Fig 3 is also redundant given that RNAseq can be considered a solid method that doesn't require verification by a second transcriptional method. I only have the following comments:

- 1) lines 30-31: "...to study the thiol-reactive mode of action of HOSCN stress in *S. aureus*". First of all, the thiol-reactive mode of HOSCN is already well studied and, in my opinion, not what the manuscript reports. More so, the study reveals how *S. aureus* responds to and defends HOSCN.
- 2) Lines 146-147: "...which leads to a half-maximal growth rate." I don't see this happening. Rather, cells remain in a concentration-dependent growth arrest after exposure to HOSCN, and then recover by growing with the same/somewhat similar growth rate. The authors acknowledge this in line 155.
- 3) Do the authors have an explanation why LB-grown *S. aureus* is more sensitive to HOSCN than RPMI-grown cells?
- 4) While Fig 2 is well illustrated, the text would benefit from more clarity. Some of the gene names are discussed without making clear what their functions are, which makes the overall readability of this section distracting.
- 5) Fig. 3: " Transcription of the HypR-controlled *merA* gene was most strongly 30-fold induced by HOSCN in Northern blots, followed by the 10-14-fold induction of the QsrR-regulated *azoR1* and *frp* transcripts." How were the fold-changes calculated? I am especially surprised about the statement concerning *azoR1*, I barely see any increase in band intensity, even though the error bar in B is very small.
- 6) Fig 4 / line 848: μM instead of mM
- 7) Fig 5A: Figure legend and Figure say 94 μM HOSCN, the text 136 μM
- 8) Fig 5C: No error bars shown. What statistical tests were performed, what is the n number etc?
- 9) Fig 7: The control for the ΔbshA strain is missing.
- 10) The upregulation of many chaperones suggest that HOSCN causes protein aggregation in *S. aureus*, which is in contrast to Ref 61, which showed that the production of polyphosphate in *P. aeruginosa* suppresses the proteotoxic effect of HOSCN. Do the authors see a HOSCN concentration-dependent effect on protein aggregation?

Staff Comments:

Preparing Revision Guidelines

Please return the manuscript within 60 days; if you cannot complete the modification within this time period, please contact me. If you do not wish to modify the manuscript and prefer to submit it to another journal, please notify me of your decision immediately so that the manuscript may be formally withdrawn from consideration by Microbiology Spectrum.

Corresponding authors may join or renew ASM membership to obtain discounts on publication fees. Need to upgrade your

membership level? Please contact Customer Service at Service@asmusa.org.

Responses # to reviewer comments:

Reviewer #1 (Comments for the Author):

Review of Spectrum03252-23: "The neutrophil oxidant hypothiocyanous acid causes a thiol-specific stress response and an oxidative shift of the bacillithiol redox potential in *Staphylococcus aureus*"

Summary:

In this thorough and informative manuscript, Loi and colleagues dissect the response of the important pathogen *Staphylococcus aureus* to the immune oxidant hypothiocyanous acid (HOSCN), an innate immune antimicrobial for which bacterial defenses have only recently begun to be characterized. They compare the impact of HOSCN and the better-studied oxidant HOCl on the transcriptome of *S. aureus*, as well as examining the role of diverse antioxidant gene systems and the low-molecular weight thiol bacillithiol in the HOSCN response. They are coming to the conclusion that in this organism the HOSCN reductase MerA is the primary defense mechanism against HOSCN, expanding on the conclusions of their previous paper in this area.

On the whole, this is an excellent paper and a valuable contribution to the literature. There are only a few points of confusion and suggestions that I have, which are detailed below.

#We are very thankful for the very positive and supportive comments of the reviewer to our manuscript and have revised the manuscript according to all critical comments of both reviewers. We have addressed each comment in the reviewer response letter in detail.

Specific Comments:

General point: This paper uses *S. aureus* strain COL, as opposed to their previous work on strain USA300 JE2. Is there an advantage to this difference? It might be useful to mention at some point in the manuscript the relevant differences between these strains.

#We thank this reviewer for this important comment about the *S. aureus* strain differences. *S. aureus* COL and USA300 JE2 are two different MRSA isolates with different virulence factors due to mobile genetic elements, which are enriched in USA300 JE2, leading to enhanced virulence properties of USA300 JE2. While COL is an archaic HA-MRSA strain of lower virulence (W.M. Shafer, et al, Infect Immun 25(3) (1979) 902-11.), USA300 JE2 is a highly virulent community-associated MRSA strain, which was cured of the two antibiotic-resistant plasmids (Nuxoll, A.S. et al., PLoS Pathog 8: e1003033.) We had all of our mutants previously constructed in COL background and aimed to test HOSCN-sensitive phenotypes of these various mutants impaired in the oxidative and electrophile stress response in this study. Construction of the *S. aureus* mutations in other backgrounds, such as USA300 JE2 is very time-consuming and difficult due to the lack of natural genetic competence. Since many of the tested COL mutants did not show phenotypes upon HOSCN stress, it would bring us not any advance to construct these many mutants in USA300 JE2 again. In addition, we aimed to test our Brx-roGFP2 biosensor upon HOSCN stress, but Brx-roGFP2 biosensor expression and fluorescence in USA300 JE2 background is lower and hence the measurements are more difficult (Loi et al., ARS 26(15):835-848.doi: 10.1089/ars.2016.6733.). Thus, we chose to make all experiments in COL background in this work due to the available mutants and better biosensor measurements. We added in the manuscript a short description of COL and USA300 JE2 backgrounds in the Methods part (Page 17) as follows:

"While previous phenotype analyses was performed with the *S. aureus* USA300 JE2 wild type, $\Delta hypR$ and $\Delta merA$ mutants, we have used *S. aureus* COL in this work, due to the availability of a large mutant collection constructed in the COL background. *S. aureus* USA300 JE2 is a

community-acquired highly virulent MRSA isolate, cured of two antibiotic resistance plasmids [1]. *S. aureus* COL is an archaic hospital-acquired MRSA isolate of lower virulence compared to USA300 JE2 [2].”

Figures 3-6: Student's t-test is not appropriate for analyzing these data. Please consult a statistician to select the most appropriate analysis, but I believe some form of ANOVA with a multiple comparisons test would be required for each of these figures.

#We are thankful to the reviewers for this important comment and have included a section on the statistics at the end of the Methods part and specified the detailed statistical analysis in each figure legends. Statistical analysis of the growth analysis, cell viability assays, expression analysis using Northern blots and biosensor measurements were calculated using the Student's unpaired two-tailed t-test for two samples with unequal variance. We always compared two samples in the Student's unpaired two-tailed t-test, such as HOSCN-treated WT versus HOSCN-treated mutants or complemented strains in growth curves, or expression in Northern blots of each gene in the WT after HOSCN stress versus the untreated control, where the control was set to 1. Similar the biosensor oxidation was analysed in HOSCN WT versus the HOSCN treated *merA* or *hypR* mutants. The same statistical tests were applied in all of our previously published studies for related growth and survival differences of mutants versus wild type, gene expression analysis and biosensor measurements, which always compared two samples with unequal variance. We have added the specific comparisons in the legends and provided the details of the comparison of the two groups.

The following sentence was included in the Methods (Page 18):

“Statistical analysis of the growth curves, survival assays, Northern blot transcription and Brx-roGFP2 biosensor measurements were performed using Student's unpaired two-tailed t-test for two samples with unequal variance by the graph prism software. The specific samples compared in the Student's unpaired two-tailed t-test are indicated in the figure legends and the *p*-values shown for each comparison.”

Lines 145-158: What is the relevance or interpretation of increased resistance to HOSCN in RPMI vs. LB? Does RPMI contain more HOSCN-reactive thiols than LB?

#We do not have any explanation for the differences in HOSCN sensitivity in RPMI versus LB medium. The composition of RPMI is very different compared to LB. RPMI medium contains defined concentrations of amino acids, salt, many vitamins, glucose and 3 μ M glutathione, but no complex peptides, proteins and lipids as found in rich LB medium. It could be possible that the GSH reacts with HOSCN and inactivates the oxidant, although the amounts of 3 μ M GSH are very low in RPMI. However, GSH uptake from the medium has been previously shown in *S. aureus* strains grown in rich TSB medium (Pöther, DC et al., 2013, Int J Med Microbiol 303, Pages 114-123), indicating that LB may also contain GSH, which is present in yeast extract. Thus, it is unclear if GSH plays a role in the different HOSCN sensitivities of *S. aureus* in RPMI vs. LB. There is also no acidification of the medium observed upon HOSCN addition in LB or RPMI medium as shown in our unpublished growth experiments, ruling out any pH effects on the sensitivity of *S. aureus* cells upon HOSCN stress. We really do not know the reasons for the high resistance towards HOSCN in RPMI medium, whereas *S. aureus* is more sensitive to HOSCN when grown in LB. This is an open question for future research.

Line 152: In this paper, 250 μ M HOSCN is referred to as a "high" HOSCN concentration. In papers published on other species (e.g. *S. pneumoniae* or *E. coli*), HOSCN doses as high as 800 μ M are common. How does the ability of *S. aureus* to resist HOSCN compare to other species? *S. pneumoniae* may be the most relevant comparison here, since they are both Gram-positive organisms encoding MerA/Har.

#We thank this reviewer for raising this point about the different HOSCN concentrations used in this and the previous paper on MerA in *S. aureus* (Shearer et al., 2023. Mol Microbiol 119, 456-470. <https://doi.org/10.1111/mmi.15035>). The assays used to test the HOSCN sensitivity (as CFUs) on different species (*S. aureus*, *Streptococcus pneumoniae*, *Pseudomonas aeruginosa*) in Fig. 1 of Shearer et al., 2023 is different compared to the growth assays in LB shown in the same manuscript (Fig. 6 of Shearer et al., 2023). For the killing assays (Fig. 1 of Shearer et al., 2023), the bacterial overnight cultures was washed and suspended in HBSS, pH 6.8, followed by OD measurements and calculation of the CFU of 2.5×10^5 CFU/mL (based on standard curves). Then the 2.5×10^5 CFU/mL were exposed to 0–800 μ M HOSCN in HBSS, pH 6.8 for up to 3 h to count CFUs after different times and concentrations. In contrast, in the growth experiments of Fig. 6 and SI-Figure of Shearer et al., 2023, the *S. aureus* cells were grown in LB and treated with 100-200 μ M HOSCN from the beginning of inoculation, leading the growth inhibition of HOSCN. In this work, we treated *S. aureus* to 176-250 μ M HOCl during the log phase at an OD of 0.5, showing also growth inhibition at similar concentrations as the previous microplate growth experiments (Fig. 6 and SI-Figure of Shearer et al., 2023). However, we did not determined the survival with 800 μ M HOSCN, since our killing assay is different from the Dickerhof lab. For our survival assays, we treat the bacterial culture during the log phase with the highest HOSCN concentration (250 μ M) as possible at an OD of 0.5 to HOSCN and plate for CFUs after 2 and 4 h (Fig. 5C this manuscript), showing that the WT survives to about 70% after 2h, but proceeds with growth after 4h. We cannot use higher HOSCN doses in our killing assay during the growth of the *S. aureus* to avoid dilution of the bacterial culture at the OD of 0.5, since the HOSCN stock is limited from the stock solution made using the LPO-SCN system. Thus, 250 μ M HOSCN was the highest possible concentration we could use in our killing assays, where no dilution of the log phase culture occurred. As for the growth sensitivity, the applied doses of 176 and 250 μ M in our work are similar as in the previous paper (Fig. 6 and SI-figures in Shearer et al., 2023).

In the introduction (Page 4), we already mentioned the different sensitivities of the 3 pathogens towards HOSCN. We added the now the specific concentrations in the buffer killing assays used for the sensitivities towards HOSCN stress in the introduction as follows:

“The bacterial responses and defense mechanisms against HOSCN stress have been investigated in important human pathogens, such as *Escherichia coli*, *Streptococcus pneumoniae*, *Pseudomonas aeruginosa* and *S. aureus* (10, 13). While *P. aeruginosa* is highly sensitive towards HOSCN stress, the Gram-positive respiratory pathogens *S. pneumoniae* and *S. aureus* were found to be much more HOSCN resistant (23, 24). In buffer killing assays, more than 50% of *S. aureus* and *S. pneumoniae* cells survived the treatment with 800 μ M HOSCN after 2 hours, whereas *P. aeruginosa* was rapidly killed within 30 min (25).”

Lines 210-214: I find the report of down-regulation as fractional values (e.g. "0.1-fold") a little confusing and counter-intuitive. I would much prefer that this same difference be expressed as a 10-fold reduction in expression instead (and similarly for all of the fractions in this section).

#We thank this reviewer for this comment. However, the reported A-values (log₂ average intensity) and M-values (log₂ fold-changes) between HOSCN versus control as shown in Tables S1-S2 of up- and downregulated transcripts are calculated by the DeSeq2 software and shown always like that in all previous transcriptome analyses, we have published. We have used the fold-changes calculated from the log₂ fold-changes (M-values) for data interpretation throughout the text. In case of negative M-values, the calculated fold-changes are calculated between 0-0.5. Thus, we cannot changes the values for down-regulated genes since these are the actual data obtained by the Deseq2 software. In addition, the values of fold-changes and their interpretation must be coherent with other transcriptome data, which we published in several previous paper (e.g. about HOCl, Lapachol, AGXX, allicin, MHQ and

itaconic acid stress). Thus, it is not possible to change the expression values of down-regulated genes.

Line 219: For readers not familiar with the stringent response, it would be useful to define that before this point.

#We agree with this suggestion of the reviewer and have shortly introduced the stringent response in the results part of the RNA-seq transcriptome changes regarding translation-associated genes (Page 8) as follows:

“The down-regulation of translation under stress and starvation conditions is associated with the synthesis of the alarmones (p)ppGpp, leading to the stringent response by inhibition of processes required for active growth, while amino acid biosynthesis and stress response pathways are activated to promote bacterial survival [3].

Line 271: I don't know if I would characterize this as "similarly", especially since the authors note in the Discussion how GSH seems to be much more important to HOSCN response in *S. pneumoniae* than BSH is in *S. aureus*.

#We agree with the reviewer and have deleted “similarly” (Page 10), since the phenotypes of *S. aureus merA* and *bshA* single and double mutants towards HOSCN stress are different compared the phenotypes of *har*, *gor* and *gshT* single and double mutants in the pneumococcus.

Lines 275-286 and Figure 5C: I do not think this qualifies as a "% survival assay" when most of the strains reported are either not dying at all or are growing more or less robustly. It might be better to simply refer to this as a viable cell assay. Do the colors of the bars have any meaning? I suggest plotting the percent cell viability on the Y-axis on a log scale, which would have the advantage of making the 100% value more obvious (a dashed line there would also help the reader tell whether any of the WT cells, for example, are being killed) and would also remove the need for a broken axis. In general, I prefer plotting individual points rather than or in addition to histogram bars, but I recognize that is a personal aesthetic choice.

#We thank this reviewer for this important comment and agree with the changes in Fig. 5C. We have removed the broken axis and plotted the values of the % viability in log scale as the bacteria grow also exponentially. We also changed survival assay in viability assay in the manuscript. However, we would prefer the bar plots with SD of the mean as shown in all previous manuscripts regarding these survival assays.

Figure 2: This figure is extremely complex, and while there may be a limit to what the authors can do about that (the data are complex!), I have a few suggestions that may help. While A-value is defined in the Methods section, it is not obvious from what's in the figure and legend. It may be better to express this in simpler terms for the non-expert; perhaps "A-value (log₂ mean gene expression)"? A definition in the figure legend is certainly necessary. Secondly, I don't think the difference between light grey and dark grey dots is explained anywhere. I think the dark grey with black borders are the SigB operon, but in general, I'm struggling with the multiple uses of grey in this figure. I recognize the challenge of finding a color scheme that works well in a figure like this, but I have found that the palettes described at <https://personal.sron.nl/~pault/> are helpful in this regard.

#We agree with the reviewer in particularly about the different grey colors and the missing explanation for the A and M value and the dark grey symbols in the legend, which was however explained in the Results section and in the detailed Tables S1 and S2, where all RNAseq data were presented in detail. However, we have now improved the figure 2 for the grey scales and understanding of A and M value. We have changed the grey color of the symbol for the “sigB

regulon” in white color with black border in the M/A scatter plot. We further explained in the legend the M and A values again in detail to facilitate the understanding of the M/A ratio/intensity scatter plot. We also explained the dark grey symbols, which are differentially expressed genes without allocation to specific regulons.

The legend of Fig. 2 was changed as follows (Page 27):

“The M value represents the log₂ fold-change and the A value is the log₂ average intensity (log₂ base mean) of each transcript under HOSCN stress versus the untreated control. Light gray symbols denote transcripts with no fold-changes ($p > 0.05$). Colored symbols and dark grey symbols indicate significantly induced or repressed transcripts (M-value ≥ 1.0 or ≤ -1.0 ; $p \leq 0.01$). The significantly up- and downregulated regulons were functionally classified into the HOSCN/ROS defense (HypR, TetR, PerR, QsrR, MhqR), proteostasis (CtsR, HrcA), sulfide detoxification (CstR), Cys biosynthesis (T-box Cys), metal stress responses (CsoR, CzrA, Fur, Zur), glucose catabolism (GapR) and virulence (SigB, AgrA) regulons, which revealed a strong thiol-specific oxidative and electrophile stress response and protein damage under HOSCN stress. These differentially expressed regulons with significant fold-changes were color coded as indicated in the legend.”

Figure 3B: I will argue again for a log-scale rather than a broken axis for this graph as well.

#We agree and have removed the broken axis in the graph of Fig. 3B. Since the SigB regulon genes *asp23* and *hchA* are downregulated, we have now included a second diagram showing only both down-regulated genes separately from the graph of induced genes. However, gene expression changes (fold changes) are usually not plotted in log scale. We never have plotted transcriptional data in log scale in any previous manuscript.

Figure 7: This is the figure I have the most trouble with in the paper. If this Western blot uses antibodies specific to protein bacillithiolation, why are there any bands detected in the $\Delta bshA$ strain? There are certainly more bands in the lanes with HOSCN, and those do appear to be eliminated in the reducing gel, but I'm very puzzled by the presence of so many dominant bands that are neither BSH- nor HOSCN-dependent. Please include more information on the interpretation of this experiment.

#We thank this reviewer for this comment about the detection of S-bacillithiolations under HOSCN stress using the non-reducing BSH specific Western blot analysis in Fig. 7. The used BSH antibody is a polyclonal rabbit antiserum, which we generated some years ago. This antibody was always successfully used to show S-bacillithiolations in many studies, but as polyclonal antibody this shows cross-reactivity to other cytoplasmic proteins. For example, upon HOCl stress, the GapDH-SSB band was detected as major S-bacillithiolated protein in *S. aureus* cells, which was identified also by mass spec previously. Thus, the BSH antibodies detects specific bands of S-bacillithiolated proteins under thiol stress, which are absent in the control and the stressed *bshA* mutant and disappear in the reducing BSH Western blot. The cross-reactivity of the BSH antibodies explains the many other bands in the wild type control and in the *bshA* mutant as well as in the reducing blots. However, the increased intensity pattern and additional bands of S-bacillithiolated proteins can be clearly detected in the WT and *merA* mutant under HOSCN stress, which was not detected in the *bshA* mutant under HOSCN stress and disappeared in the reducing Western blots. This cross-reactivity has been noted in earlier studies (Imber et al., ARS 2018; Linzner et al., Front Microbiology 2019). This information on the cross-reactivity of the polyclonal BSH antiserum was added now in the manuscript results part (Page 12-13) as follows:

“As noted earlier, the polyclonal BSH rabbit antiserum shows cross-reactivity with abundant cellular proteins [4], detected as background in the untreated WT and the $\Delta merA$ mutant as well as in the HOSCN-treated *bshA* mutant. However, increased levels of specific bands of S-

bacillithiolated proteins were reproducibly detected in the WT and the $\Delta merA$ mutant under HOSCN stress. The specific targets for reversible thiol-oxidation including S-bacillithiolations under HOSCN stress will be elucidated using quantitative redox proteomics approaches in our future research.”

Reviewer #2 (Comments for the Author):

In the current manuscript, the authors follow up on their previous findings and investigate how the gram-positive bacterium *S. aureus* responds to hypothiocyanous acid stress using transcriptomic, phenotypic, and biosensor approaches. The manuscript is well-written and easy to follow. While the growth curve analyses of $\Delta merA$ and $\Delta hypR$ strains during HOSCN stress are not novel (published in Ref 29 and acknowledged by the authors in lines 101-105), a transcriptomic response and the cellular oxidation status have not been reported before. Fig 3 is also redundant given that RNAseq can be considered a solid method that doesn't require verification by a second transcriptional method.

#We thank this reviewer for the positive and critical comments to our manuscript. We are aware that the phenotypes of the *merA* and *hypR* mutants have been published in the USA300 JE2 background (Shearer et al., 2023), but not in the COL as in this work. Thus, all presented data of this study in the COL background are novel and not yet published, which is clearly stated in the text. And in this work, we constructed a lot of other mutants in single and double with the *merA* deletion to identify further defense mechanisms against HOSCN as another important goal of this study. As for Fig. 3, we think it is important to approve at least the most important hits of the RNAseq data using Northern blot analysis, as this is often requested to confirm the transcriptome analyses. Thus, Fig. 3 will remain in the manuscript to confirm the RNAseq data.

I only have the following comments:

1) lines 30-31: "...to study the thiol-reactive mode of action of HOSCN stress in *S. aureus*".

First of all, the thiol-reactive mode of HOSCN is already well studied and, in my opinion, not what the manuscript reports. More so, the study reveals how *S. aureus* responds to and defends HOSCN.

#We thank this reviewer for raising this point and have corrected the statement about the response of *S. aureus* towards HOSCN stress as suggested as follows in the Abstract Page 2:

“In this work, we applied RNAseq transcriptomics, Brx-roGFP2 biosensor measurements and phenotype analyses to investigate the stress responses and defense mechanisms of *S. aureus* towards HOSCN stress.”

2) Lines 146-147: "...which leads to a half-maximal growth rate." I don't see this happening. Rather, cells remain in a concentration-dependent growth arrest after exposure to HOSCN, and then recover by growing with the same/somewhat similar growth rate. The authors acknowledge this in line 155.

#We thank the reviewer for this comments and agree that HOSCN rather impairs the growth. We corrected this statement as follows (Page 6):

“First, we determined the sub-lethal concentration of HOSCN, which impairs the growth of *S. aureus* COL, when cultivated in LB and RPMI medium, without killing effects.”

3) Do the authors have an explanation why LB-grown *S. aureus* is more sensitive to HOISCN than RPMI-grown cells?

#We do not have any explanation for the differences in HOSCN sensitivity in RPMI versus LB medium. The composition of RPMI is very different compared to LB. RPMI medium contains defined concentrations of amino acids, salt, many vitamins, glucose and 3 μ M glutathione, but no complex peptides, proteins and lipids as found in rich LB medium. It could be possible that the GSH reacts with HOSCN and inactivates the oxidant, although the amounts of 3 μ M GSH are very low in RPMI. However, GSH uptake from the medium has been previously shown in *S. aureus* strains grown in rich TSB medium (Pöther, DC et al., 2013, Int J Med Microbiol 303, Pages 114-123), indicating that LB may also contain GSH, which is present in yeast extract. Thus, it is unclear if GSH plays a role in the different HOSCN sensitivities of *S. aureus* in RPMI vs. LB. There is also no acidification of the medium observed upon HOSCN in LB or RPMI medium as shown in our unpublished growth experiments, ruling out any pH effects on the sensitivity of *S. aureus* cells upon HOSCN stress. We really do not know what are the reasons for the high resistance towards HOSCN in RPMI medium, whereas *S. aureus* is more sensitive to HOSCN when grown in LB. This is an open question for future research.

4) While Fig 2 is well illustrated, the text would benefit from more clarity. Some of the gene names are discussed without making clear what their functions are, which makes the overall readability of this section distracting.

#We thank this reviewer for this comment. We agree and have revised the transcriptome results part carefully to explain the functions of the discussed genes more clearly to improve the understanding.

The following genes functions of have been revised on Page 7-9:

"In addition, the quinone-sensing QsrR and MhqR regulons respond strongly (7.5-38.7-fold) to HOSCN stress, including the *catE-SACOL0409-azoR1*, *yodC*, *catE2*, *frp* and *mhqRED* operons, which functions in detoxification of quinones and confer resistance towards quinones, antibiotics and oxidants (48, 49).

Additionally, HOSCN upregulates transcription of the reactive sulfur species (RSS)-sensing CstR regulon, comprising the *cstAB-sqr* and *cstR-tauE* operons (3-33-fold), which encode the multidomain sulfurtransferase (CstA), persulfide dioxygenase-sulfurtransferase (CstB) and sulfide:quinone oxidoreductase (Sqr) involved in detoxification of hydrogen sulfide in *S. aureus* [5-7].

Furthermore, the large GraRS cell wall stress regulon and the virulence regulons controlled by the accessory gene regulator A (AgrA) and the general stress and starvation sigma factor B (SigB) were differentially transcribed under HOSCN stress.

The down-regulation of translation under stress and starvation conditions is associated with the synthesis of the alarmones (p)ppGpp, leading to the stringent response by inhibition of processes required for active growth, while amino acid biosynthesis and stress response pathways are activated to promote bacterial survival [3]."

5) Fig. 3: " Transcription of the HypR-controlled *merA* gene was most strongly 30-fold induced by HOSCN in Northern blots, followed by the 10-14-fold induction of the QsrR-regulated *azoR1* and *frp* transcripts." How were the fold-changes calculated? I am especially surprised about the statement concerning *azoR1*, I barely see any increase in band intensity, even though the error bar in B is very small.

#We thank the reviewer for this comment about the quantification of the Northern blot and the weak *azoR1* transcript in Fig. 3. The calculation of the fold changes is described now more detailed in the figure 3B, C legend as follows:

“Fig. 3 B, C) Quantification of the transcriptional induction of the genes after HOSCN stress in *S. aureus* was performed from the Northern blot images using ImageJ. HOSCN-induced fold-changes were calculated from 3 biological replicates and error bars represent the SD. For calculation of the fold-changes after HOSCN stress, the transcript intensities of each gene after HOSCN stress were normalized to the mRNA intensity of the untreated control, which was set to 1. The statistics of the fold-changes for each gene under HOSCN stress versus the control were calculated using a Student’s unpaired two-tailed *t*-test for two samples with unequal variance by the graph prism software.”

The *azoR1* transcript had a low intensity after HOSCN stress, but no basal transcript was detected in the untreated control, which explains the high fold-change of *azoR1* transcription upon HOSCN stress. We quantified the transcripts from Northern blots of 3 bioreplicates, which showed similar fold-changes for the *azoR1* transcription explaining the low error bars. However, to better visualize *azoR1* transcription, we now show the image of another bioreplicate in Fig. 3A, where the *azoR1* band can be better detected, but the background staining was darker in this Northern blot.

6) Fig 4 / line 848: uM instead of mM

#This error has been corrected in μM as suggested.

7) Fig 5A: Figure legend and Figure say 94 uM HOSCN, the text 136 uM

#We corrected 94 and 176 μM HOSCN in the text to be consistent with the figure 5A.

8) Fig 5C: No error bars shown. What statistical tests were performed, what is the n number etc?

#The statistics was calculated and the errors bars shown in Fig. 5C, but these were very small and hard to see on the previous TIF image. In the revised version, we show the % viability in log scale as suggested by Rev1. Thus, the error bars are better visualized now in the revised log-scale graph.

The legend of Fig. 5C was also updated with the more detailed information of the statistics and replicates and errors bars as follows (Page 30/31):

“Fig. 5 C) For viability assays, the *S. aureus* WT, $\Delta bshA$, $\Delta merA$ and $\Delta merA\Delta bshA$ mutants and the *merA*⁺ complemented strain were treated with 250 μM HOSCN during the log phase and plated for CFUs after 2 and 4 hours of stress exposure. The percentage viability rate was calculated in relation to that of the untreated WT, which was set to 100%. Mean values were calculated from 3 biological replicates and error bars represent the SD. The statistics of viability differences were calculated between the HOSCN-treated mutants or complemented strains versus the HOSCN treated WT as well as between the HOSCN-treated $\Delta merA\Delta bshA$ double mutant versus the $\Delta merA$ or $\Delta bshA$ mutants using a Student’s unpaired two-tailed *t*-test for two samples with unequal variance by the graph prism software. Symbols are: ^{ns} $p > 0.05$, * $p \leq 0.05$, ** $p \leq 0.01$ and *** $p \leq 0.001$.”

9) Fig 7: The control for the $\Delta bshA$ strain is missing.

#Due to the limited space of the 10 lanes in the BSH-specific Western blot, the *bshA* mutant control sample was left out from this blot, because there is no difference between the *bshA* mutant control and under HOSCN stress, since this strain does not produce BSH and therefore no bacillithiolations are detected. However, we had done many previous Western blot experiments confirming no difference between the *bshA* control and under HOCSN stress. We send one representative image as **Figure R1** for the reviewer confirming no difference between the bands in the *bshA* control and under HOCSN stress.

10) The upregulation of many chaperones suggest that HOSCN causes protein aggregation in *S. aureus*, which is in contrast to Ref 61, which showed that the production of polyphosphate in *P. aeruginosa* suppresses the proteotoxic effect of HOSCN. Do the authors see a HOSCN concentration-dependent effect on protein aggregation?

#We thank this reviewer for raising this point about our suggested protein aggregation upon HOSCN stress in *S. aureus* due to the strong up-regulation of the chaperones and proteases of the CtsR and HrcA regulons. We performed now the experiments to isolate intracellular protein aggregates according to the previously described protocols [8, 9]. However, we could not detect increased protein aggregation upon HOSCN, which is in agreement with previous reports in *P. aeruginosa* under sublethal HOSCN (Fig. S4). We thus corrected our statement in the text and the summary figure 8 to avoid the term “protein aggregates”. We now explain that the chaperones and proteases are strongly induced under HOSCN stress due to increased protein thiol-oxidation and protein unfolding.

We also added in the discussion the following explanation (Page 15):

“However, while protein aggregation is caused by HOCl stress as bacterial killing mechanism [10-12], we did not observed increased formation of protein aggregates in *S. aureus* after the exposure to 176 and 250 μ M HOSCN stress (Fig. S4), which is consistent with the results obtained in *P. aeruginosa* WT cells after treatment with sub-lethal HOSCN stress [8].”

References:

- [1] A.S. Nuxoll, S.M. Halouska, M.R. Sadykov, M.L. Hanke, K.W. Bayles, T. Kielian, R. Powers, P.D. Fey, CcpA regulates arginine biosynthesis in *Staphylococcus aureus* through repression of proline catabolism, *PLoS Pathog* 8(11) (2012) e1003033.
- [2] W.M. Shafer, J.J. Iandolo, Genetics of staphylococcal enterotoxin B in methicillin-resistant isolates of *Staphylococcus aureus*, *Infect Immun* 25(3) (1979) 902-11.
- [3] S.E. Irving, N.R. Choudhury, R.M. Corrigan, The stringent response and physiological roles of (pp)pGpp in bacteria, *Nat Rev Microbiol* 19(4) (2021) 256-271.
- [4] M. Imber, N.T.T. Huyen, A.J. Pietrzyk-Brzezinska, V.V. Loi, M. Hillion, J. Bernhardt, L. Thärichen, K. Kolsek, M. Saleh, C.J. Hamilton, L. Adrian, F. Gräter, M.C. Wahl, H. Antelmann, Protein S-bacillithiolation functions in thiol protection and redox regulation of the glyceraldehyde-3-phosphate dehydrogenase Gap in *Staphylococcus aureus* under hypochlorite stress, *Antioxid Redox Signal* 28(6) (2018) 410-430.
- [5] J. Shen, M.E. Keithly, R.N. Armstrong, K.A. Higgins, K.A. Edmonds, D.P. Giedroc, *Staphylococcus aureus* CstB is a novel multidomain persulfide dioxygenase-sulfurtransferase involved in hydrogen sulfide detoxification, *Biochemistry* 54(29) (2015) 4542-54.
- [6] K.A. Higgins, H. Peng, J.L. Luebke, F.M. Chang, D.P. Giedroc, Conformational analysis and chemical reactivity of the multidomain sulfurtransferase, *Staphylococcus aureus* CstA, *Biochemistry* 54(14) (2015) 2385-98.
- [7] J. Shen, H. Peng, Y. Zhang, J.C. Trinidad, D.P. Giedroc, *Staphylococcus aureus* *sqr* encodes a type II sulfide:quinone oxidoreductase and impacts reactive sulfur speciation in cells, *Biochemistry* 55(47) (2016) 6524-6534.
- [8] B. Groitl, J.U. Dahl, J.W. Schroeder, U. Jakob, *Pseudomonas aeruginosa* defense systems against microbicidal oxidants, *Mol Microbiol* 106(3) (2017) 335-350.
- [9] T. Tomoyasu, F. Arsene, T. Ogura, B. Bukau, The C terminus of sigma(32) is not essential for degradation by FtsH, *J Bacteriol* 183(20) (2001) 5911-7.
- [10] M.J. Gray, W.Y. Wholey, U. Jakob, Bacterial responses to reactive chlorine species, *Annu Rev Microbiol* (2013).
- [11] J. Winter, M. Ilbert, P.C. Graf, D. Ozcelik, U. Jakob, Bleach activates a redox-regulated chaperone by oxidative protein unfolding, *Cell* 135(4) (2008) 691-701.

[12] A. Ulfig, L.I. Leichert, The effects of neutrophil-generated hypochlorous acid and other hypohalous acids on host and pathogens, *Cell Mol Life Sci* 78(2) (2021) 385-414.

October 2, 2023

Prof. Haike Antelmann
Freie Universität Berlin
Institute of Biology-Microbiology
Königin-Luise-Str. 12-16
Berlin D-14195
Germany

Re: Spectrum03252-23R1 (The neutrophil oxidant hypochlorous acid causes a thiol-specific stress response and an oxidative shift of the bacillithiol redox potential in *Staphylococcus aureus*)

Dear Prof. Haike Antelmann:

Your manuscript has been accepted, and I am forwarding it to the ASM Journals Department for publication. You will be notified when your proofs are ready to be viewed.

Sincerely,

Artem Rogovskyy
Editor, Microbiology Spectrum
